# A novel eukaryotic RdRP-dependent small RNA pathway represses antiviral immunity by controlling an ERK pathway component in the black-legged tick

Canran Feng[1], Kyosuke Torimaru[1], Mandy Yu Theng Lim[2,3], Li-Ling Chak[2], Masami Shiimori[1], Kosuke Tsuji[1], Tetsuya Tanaka[4], Junko Iida[1], Katsutomo Okamura[1,2,3]*

1 Nara Institute of Science and Technology, Nara, Japan, 2 Temasek Life Sciences Laboratory, 1 Research Link, National University of Singapore, Singapore, Singapore, 3 School of Biological Sciences, Nanyang Technological University, Singapore, Singapore, 4 Joint Faculty of Veterinary Medicine, Laboratory of Infectious Diseases, Kagoshima University, Kagoshima, Japan

* okamurak@bs.naist.jp

**Data Availability Statement:** The small RNA library data produced for this study are deposited at NCBI SRA under GSE183810. The UCSC

## Abstract

Small regulatory RNAs (sRNAs) are involved in antiviral defense and gene regulation. Although roles of RNA-dependent RNA Polymerases (RdRPs) in sRNA biology are extensively studied in nematodes, plants and fungi, understanding of RdRP homologs in other animals is still lacking. Here, we study sRNAs in the ISE6 cell line, which is derived from the black-legged tick, an important vector of human and animal pathogens. We find abundant classes of ~22nt sRNAs that require specific combinations of RdRPs and sRNA effector proteins (Argonautes or AGOs). RdRP1-dependent sRNAs possess 5'-monophosphates and are mainly derived from RNA polymerase III-transcribed genes and repetitive elements. Knockdown of some RdRP homologs misregulates genes including RNAi-related genes and the regulator of immune response Dsor1. Sensor assays demonstrate that Dsor1 is downregulated by RdRP1 through the 3'UTR that contains a target site of RdRP1-dependent repeat-derived sRNAs. Consistent with viral gene repression by the RNAi mechanism using virus-derived small interfering RNAs, viral transcripts are upregulated by AGO knockdown. On the other hand, RdRP1 knockdown unexpectedly results in downregulation of viral transcripts. This effect is dependent on Dsor1, suggesting that antiviral immunity is enhanced by RdRP1 knockdown through Dsor1 upregulation. We propose that tick sRNA pathways control multiple aspects of immune response via RNAi and regulation of signaling pathways.

## Introduction

Foreign nucleic acids such as phages and viruses pose constant threats to host cells. To inactivate invading agents, cells are equipped with defense mechanisms that use short fragments of nucleic acids to silence those foreign nucleic acids [1].

assembly hubs with the RNAseq mapping tracks are available at https://data.cyverse.org/dav-anon/iplant/home/okamuralab/trackhub/Isc_ISE6/Iscal1_hub.txt and https://data.cyverse.org/dav-anon/iplant/home/okamuralab/trackhub/ucscgb_haeL2018/hubHaeL2018.txt.

**Funding:** Research in K.O.'s group was supported by the National Research Foundation, Prime Minister's Office, Singapore under its NRF Fellowship Programme (NRF2011NRF-NRFF001-042), Temasek Life Sciences Laboratory core funding and the JSPS Fund for the Promotion of Joint International Research (Returning Researcher Development Research, 17K20145). Work in the T. T.'s group was supported by Takeda Science Foundation. The content is solely the responsibility of the authors and does not necessarily represent the official views of these agencies. There was no additional external funding received for this study.

**Competing interests:** The authors have declared that no competing interests exist.

In prokaryotes, a common defense mechanism involving CRISPR (Clustered Regularly Interspaced Short Palindromic Repeats) produces short RNAs that bind to the effector protein typically to degrade foreign DNA or RNA in a sequence-specific manner [2]. Another major class of defense systems using small RNAs (sRNAs) involves another family of sRNA binding proteins known as Argonautes (AGOs). The AGO system is widely conserved in various organisms and can be found in both eukaryotes and prokaryotes [3].

The large diversity of sRNA pathways is assumed to be a result of a relentless arms race between the host cell and the invading nucleic acids. In bilaterian animals, three distinct AGO-dependent sRNA pathways, namely the PIWI-interacting RNA (piRNA), small interfering RNA (siRNA) and microRNA (miRNA) pathways, are widely present [4]. These pathways were initially characterized in a few model animals including *Drosophila*, *C. elegans* and mice [5]. Expression of the piRNA pathway components is virtually restricted within gonads in these organisms and they appear to be specialized in silencing transposons while also playing roles in gene regulation [6,7]. However, recent studies argue against the notion that active piRNA production is generally confined in gonads as abundant piRNAs have been detected in somatic tissues in many arthropods, even in some insects [8–12].

The siRNA pathway is believed to be a major mechanism to control viruses in insects [13]. Mutants of the core siRNA factors exhibit elevated sensitivity to various viruses [14] and siRNAs against the virus are efficiently produced in infected cells via a specific processing mechanism [15,16]. Interestingly, the siRNA factors are among the most rapidly evolving genes potentially because they must catch up with the rapidly changing viruses. Due to the rapid evolution, the siRNA pathway shows significant evolutionary diversity [17,18]. This was first recognized by a comparison between the insect and nematode RNAi mechanisms. The clearest difference is the essential involvement of RNA-dependent RNA polymerases (RdRPs) in worms but not in flies for robust RNA silencing [19–21]. RdRPs are known to produce various sRNAs during RNA silencing in fungi and plants [22,23] whereas the gene family was believed to have been lost in the animal lineage until genome sequence analyses of non-model organisms started to discover RdRP genes in a broad range of animals [18,24,25].

The sequences of most RdRP genes support that the genes had been vertically transferred and were not introduced to the animals by the horizontal transfer, suggesting that RdRPs may have conserved biological roles in those lineages [26]. The findings that the organisms with no piRNA pathway genes always retain the RdRP genes have led to the notion that the ping-pong amplification mechanism in the piRNA pathway and RdRP-dependent sRNA production pathway have overlapping roles in reinforcing silencing activity against transposable elements (TEs) [8–11]. In nematodes, primary siRNAs generated by Dicer can trigger the generation of secondary siRNAs by recruiting an RdRP on target RNAs. Worm-like secondary sRNAs have features such as 5'-triphospophate modification and 5'-purine enrichment with their sRNA populations [26]. A past study analyzed sRNAs in another RdRP-positive lineage cephalochordates, but there was no evidence for the production of worm-like secondary sRNAs [26]. Putative RdRP genes were identified by screening *I. scapularis* genome reads [24], but their functionality in the sRNA pathway has not been examined. Experimental evidence for the functional involvement of RdRPs in sRNA biogenesis in animals outside of Nematoda is currently lacking.

To directly test if RdRPs are involved in the production of sRNAs in arachnids, we use a cell line from the model tick *Ixodes scapularis* and provide experimental evidence that abundant classes of RdRP-dependent sRNAs regulate the expression of genes in tick cells. There are distinct classes of sRNAs produced through the activity of at least two different RdRP genes. We further demonstrate the involvement of RdRP-dependent sRNAs in regulation of genes including the ERK pathway component Dsor1. Knockdown of one of the RdRPs unexpectedly

resulted in a reduction of specific viral transcripts in a Dsor1-dependent manner, suggesting that the RdRP allows for viral gene expression potentially by regulating the host's immune response by lowering the Dsor1 level. In summary, tick RdRPs are essential for the biogenesis of specific sRNAs, play roles in gene regulation and controlling viral transcript levels. This study unveils previously overlooked pathways that are potentially broadly conserved in ticks.

## Results

### RNAi factors are diversified in the *Ixodes* genome

The broad presence of recognizable RdRP genes places arachnids in a unique position in the arthropod lineage [11]. We hypothesized that arachnids might have previously unrecognized sRNA pathways fueled by the enzymatic activity of RdRPs to produce antisense RNA molecules.

To identify RdRP genes expressed in ISE6 cells, we first performed transcriptome assembly by sequencing ribosomal RNA-depleted RNAseq libraries (S1 Table, Sheet7). In the assembled transcriptome, we found 3 genes that were similar to *C. elegans* RdRP genes (S1 Fig, Gene IDs and contig sequences are on S1 Table, Sheet4). These sequences were predicted to contain the entire sequence of the conserved RdRP domain, suggesting that they were genuine RdRP genes (Fig 1A and 1B). We named them RdRP1, 3 and 4. Two (RdRP1 and RdRP3) out of the three RdRP genes were expressed at >10 TPM in our transcriptome data (Fig 1C), and we characterized these two RdRPs in the present study.

sRNA pathways often employ specific AGO proteins as their effectors [27]. We identified 6 contigs that correspond to AGO genes in the ISE6 transcriptome (Figs 1A and S2). We verified the expression of these genes by qPCR and tested the specificity of qPCR primers by introducing dsRNAs against cognate genes from regions that did not overlap with the qPCR amplicons (S3A Fig). In all cases, we observed a strong (40–90%) reduction of the tested genes upon knockdown, confirming the specificity of the qPCR assay. Therefore, we concluded that these tested genes were indeed expressed in ISE6 cells.

Among the tested genes, we found a contig in our transcriptome data that matched two annotated genes (ISCI012408 and ISCI004800). This contig was potentially derived from two genomic loci with ISCI004800(Ago3-2) sequence matched to its first part and ISCI012408 (Ago3-1) sequence matched to its second part. (S3B Fig). This contig showed the highest similarity to the *Drosophila* AGO (dAGO3) (Fig 1A), and raised a possibility that these two ISCI entries represented fragments of a dAGO3 homolog. We consider the two ISCI entries a single gene throughout the manuscript and renamed this AGO3 because knockdown using dsRNAs derived from either of the annotated sequences resulted in depletion of both ISCI012408 and ISCI004800 (S3C Fig). We found another gene belonging to the PIWI-clade (Aub). Other genes were similar to AGO-clade Argonautes, which were identified in a previous study [28]; an ortholog of the miRNA-class AGO (Ago-78) and three other genes that were relatively distant from miRNA AGOs (Ago-16, Ago-30 and Ago-96) (Fig 1A). The predicted protein sequences of Ago-16, Ago-30 and Ago-96 contained the entire PIWI domain (Fig 1B) and their catalytic residues were also conserved (S2 Fig), suggesting that they were functional slicer enzymes [29].

Using our total RNAseq data, the expression levels of these AGO genes in ISE6 cells were determined (Fig 1C). The PIWI-related genes (AGO3 and Aub) were highly expressed (>50 and >90 TPM, respectively) in ISE6 cells, which were assumed to be derived from the neural lineage [30]. Although PIWI proteins were previously believed to be confined in the animal gonad in general [31], our observation was consistent with the recent findings that piRNA mechanisms are active broadly in arachnid somatic tissues [8,11]. Expression of the PIWI

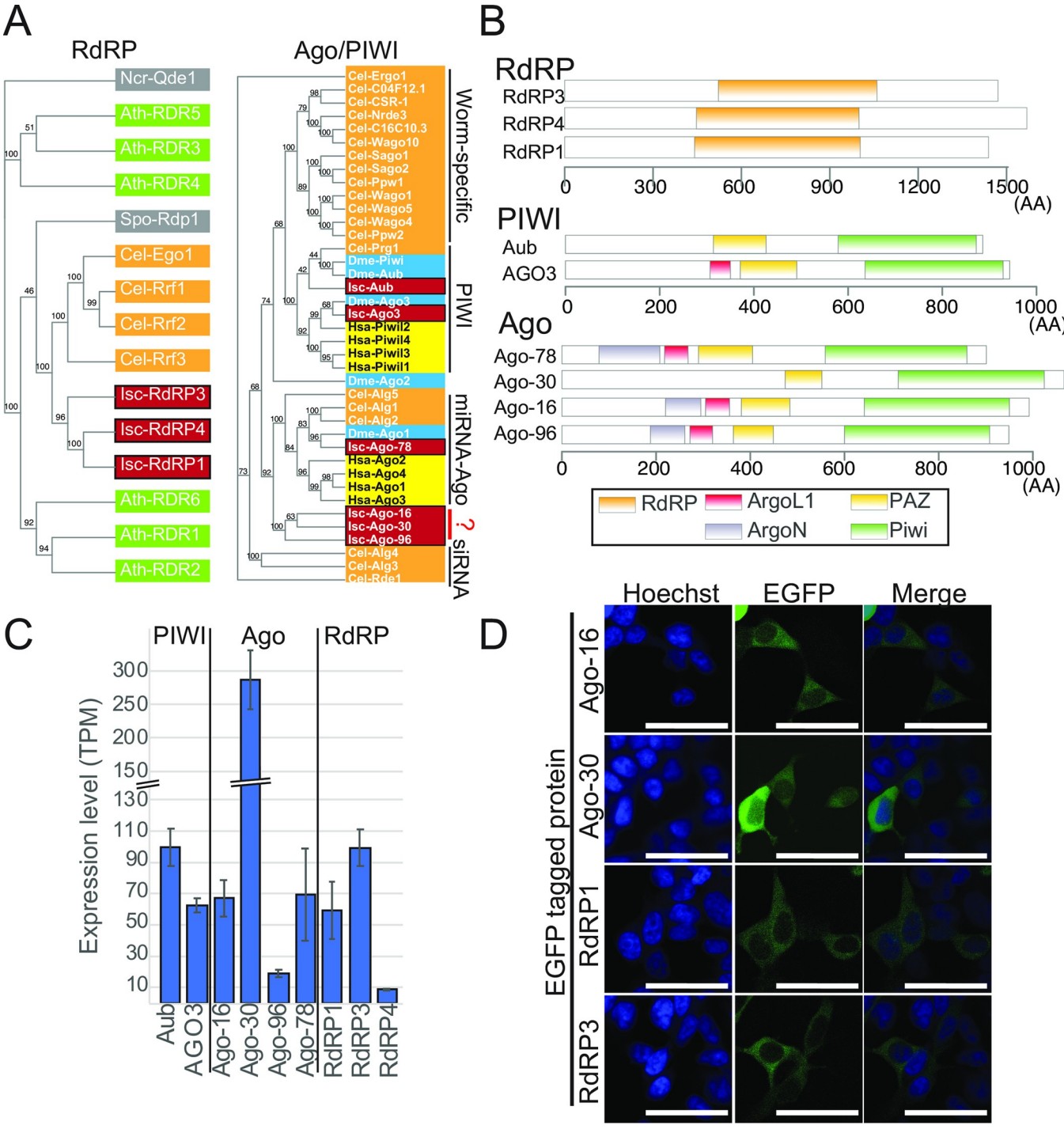

**Fig 1. Characterization of *Ixodes* RNAi factors.** (A) Phylogenetic analysis of RdRP and AGO genes. The sequences of AGO and RdRP homologs from representative organisms (*Ixodes*, *C. elegans*, *Arabidopsis thaliana*, *Neurospora crassa* and *Schizosaccharomyces pombe* for RdRP; *Ixodes*, *Drosophila*, *C. elegans* and *human* for Ago/PIWI) were aligned using MUSCLE. Gene names are surrounded by colored rectangles depending on the species (Fungi: gray, *A. thaliana*: green, *C. elegans*: orange, *Drosophila*: blue, human: yellow, *Ixodes*: dark red). (B) Protein domains found in *I. scapularis* RdRP, PIWI and AGO genes. The domains were identified using the CD-search tool (NIH/NLM/NCBI). (C) Expression of tick RNAi factors analyzed by RNAseq. The dataset of ISE6 total RNAseq from the control sample (dsGFP transfection) was used to determine the expression levels of the indicated genes. Expression levels are shown as TPM and the averages and standard deviations are shown in the bar chart (n = 3). (D) Subcellular localization of tick AGOs and RdRPs in HEK293T cells. HEK293T cells were transfected with the indicated plasmids and fixed. The cells were stained with Hoechst and observed by confocal microscopy. The bars indicate 50um.

proteins was further confirmed by Western blotting using antibodies against asymmetric di-methyl-arginine, a post-translational modification that is conserved among PIWI proteins [32]. The dsRNA against Aub decreased the signal, suggesting that IscAub was the major PIWI protein modified with asymmetric di-methyl-arginine in ISE6 cells (S3D Fig). The other four AGO genes were also highly expressed (10–250 TPM, Fig 1C). Therefore, our RNAseq data indicated that components of multiple sRNA pathways including distinct PIWI/AGO genes as well as RdRP genes were expressed in ISE6 cells.

We next studied the localization of the homologs of AGOs and RdRPs, we cloned the puta-tive ORFs of Ago-16, Ago-30, RdRP1 and RdRP3 into a mammalian expression vector with an N-terminal EGFP tag. Using the plasmids, we transfected HEK293T cells and analyzed their subcellular localization by confocal microscopy. Successful expression of the fusion proteins was confirmed by Western blotting analysis (S3E Fig). The fluorescent signals in transfected cells were detected mainly in the cytoplasm for all of the RdRP/AGO constructs (Fig 1D). Although such a heterologous experimental system might not accurately reflect their natural subcellular localization as seen with mislocalization of PIWI proteins expressed in cells lacking an active piRNA processing pathway [33], our results suggested that these proteins could local-ize in the cytoplasm at least under certain conditions.

## The sRNA repertoire of ISE6 cells

To understand the tick sRNA repertoires, we performed sRNAseq analysis (S1 Table). To understand their biogenesis mechanisms, we also generated sRNA libraries from ISE6 cells after knocking down the AGO/PIWI/RdRP genes, and each of the libraries yielded ~13–20 million reads that could be mapped to the ISE6 genome (S1 Table). sRNA reads in these librar-ies showed a bimodal distribution with peaks at ~22nt and in the 26-29nt range, which typi-cally represents peaks of miRNAs/siRNAs and piRNAs, respectively (S4A Fig).

To find clues to their functions and biogenesis mechanisms, we categorized sRNA reads based on their genomic origins (Fig 2A). The sRNA reads were sequentially mapped to the ref-erence sequences in different categories, including miRNAs, RNA polymerase III (RNAP III) transcribed genes, rRNAs, snoRNAs, protein-coding genes and repetitive sequences (See S1 Table sheet2 for the details of the reference sequences). In the control library transfected with GFP dsRNA, ~15% of the library was comprised of annotated miRNAs in miRBase (ver 22). As this class of sRNAs was strongly reduced upon the knockdown of Ago-78, which encoded the miRNA AGO ortholog (Fig 2A), this result confirmed the major role of Ago-78 in the miRNA pathway. We did not observe strong effects on miRNAs when other AGOs were knocked down (Figs 2A and S4B), suggesting that other AGOs might support functions of other sRNA classes. Repetitive sequences produced multiple classes of sRNAs. More than 40% of 22nt and 25-30nt species were derived from repeats (S4C Fig), and they might represent repeat-associated siRNAs and piRNAs as seen in the fly system [5]. Indeed, the 25-30nt species derived from repeats were strongly decreased when PIWI genes were knocked down (S4D Fig, 25-30nt). On the other hand, the 22nt species showed no strong reduction upon knockdown of any of the factors (S4D Fig, 22nt), suggesting that there were multiple classes of repeat-asso-ciated 22nt sRNAs depending on distinct biogenesis mechanisms as discussed later.

We also found a group of abundant sRNA reads derived from various genes that were known to be transcribed by RNAP III such as SRP RNA, RNase P, RNase MRP and tRNAs (Fig 2A). The read counts of sRNAs in this category accounted for ~9% of the control library, which was nearly as abundant as miRNAs (~15%) as a class. sRNAs in the "RNAP III" group in the RdRP1-KD sample were strongly reduced, suggesting that this category included abun-dant RdRP1-dependent sRNAs (Fig 2A). The sRNAs were mapped to both sense and antisense

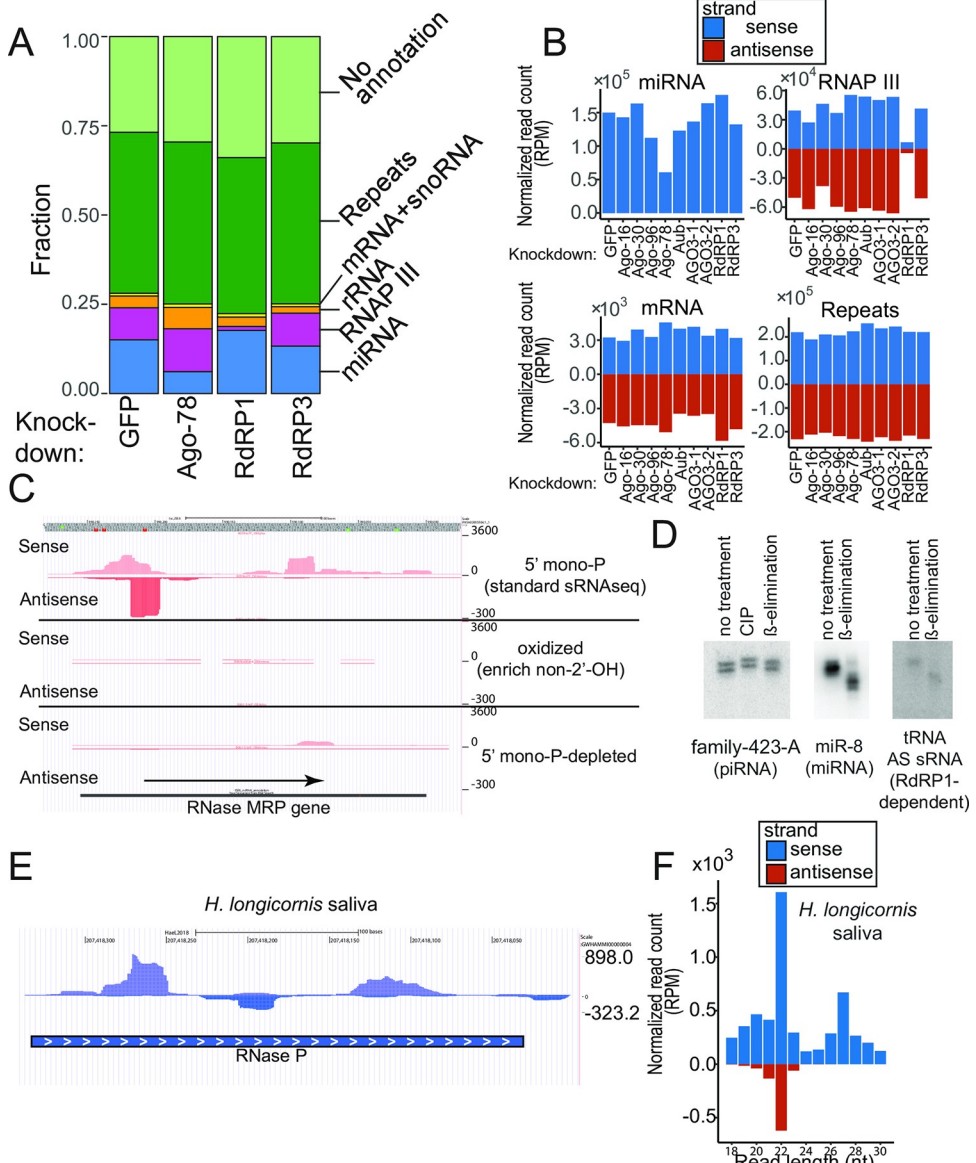

**Fig 2. Characterization of endogenous sRNA populations in ISE6 cells.** (A) Genomic origins of sRNA populations in ISE6 cells. The fractions of sRNAs derived from miRNAs (blue), RNAP III transcribed genes (pink), rRNAs (orange), snoRNAs and mRNAs (yellow) and repetitive sequences (green) are shown. sRNA reads whose genomic loci lack a gene annotation are shown as "No annotation" (light green). The percentage of reads falling within each category was calculated by dividing the number of reads in the category by the number of reads mapping to the ISE6 genomic sequence. (B) Normalized read count (RPM) of sense and antisense reads that mapped to the reference sequences of miRNAs, RNAP III transcripts, mRNAs and repeats in the indicated knockdown libraries are shown in blue and red, respectively. Negative values were given to antisense read counts. (C) Example of sRNAs in the "RNAP III" category. A UCSC screenshot of the locus of the RNAP III transcribed gene, RNase MRP RNA gene is shown, with y-axis representing the normalized read counts. On the track of "standard sRNAseq library", reads corresponding to an abundant 22nt species are mapped on the antisense strand of the RNase MRP RNA locus, and there are reads throughout both sense and antisense strands of this locus. To analyze chemical structures at their 3' nucleotides, we generated sRNA libraries after oxidizing RNA samples to enrich for sRNA species containing chemical modifications at the 2'-position. Nucleotides with free 2'-, 3'-OH groups react with periodate to form dialdehydes, which are not compatible with the 3'-linker ligation. Therefore, species with 2'-O-me modification will be overrepresented in the oxidized library. sRNAs from RNAP III-transcribed genes tended to be depleted in the oxidized library suggestive of the presence of vicinal hydroxyl groups at the 3'-nucleotide of the sRNAs. The phosphorylation status of the 5'-end could also be analyzed by taking advantage of the 5'-mono-P specificity of T4 RNA ligase, which is used for library construction. For standard sRNA library construction, the 5'-linker ligation step strongly enriches for 5'-mono-P

species. For the "5'-tri-P" library, sRNAs with 5'-di-P or 5'-tri-P groups were enriched by the removal of sRNAs with 5'-mono-P and 5'-OH groups by terminator exonuclease (See Materials and Methods). No enrichment was seen with the sRNA species from RNAP III-dependent loci in this library, suggesting that the sRNAs were 5'-mono-phosphorylated. (D) 5'- and 3'-states of a piRNA (TE- family-423-A), an antisense sRNA from a RNAP III dependent gene (tRNA ISCW004624) and a miRNA (miR-8) were verified by Northern blotting. The removal of phosphate groups at the 5'-end causes a delay in the migration on the gel. The alkaline treatment removes the oxidized RNA with dialdehydes at the 3' ends, resulting in a faster migration of the RNA on the gel (ß-elimination). (E, F) Antisense sRNAs are produced from the RNAP III-dependent gene RNase P in Asian longhorned ticks. sRNA mapping at the RNase P locus is shown (E), with y-axis representing the normalized read counts. The sRNA library was made using RNA samples extracted from purified extracellular vesicles in saliva. The size distribution of sRNAs mapping to the representative RNAP III-dependent genes (RNase P, RNase MRP and SRP RNA) are shown. RPM normalized sense and antisense read counts are shown in blue or red bars, respectively. Note that strong peaks at 22nt were observed in saliva and tick animals at different developmental stages at least on the antisense strand, indicating the conservation of an sRNA production mechanism similar to those observed in ISE6 cells (Also see S6 Fig).

directions with respect to the direction of transcription of their host genes, excluding the possibility that they were mere degradation products of abundant RNAP III transcripts (Fig 2B). Furthermore, these sRNAs were virtually eliminated when RdRP1 was knocked down, and their dependence on RdRP1 was verified by Northern blotting (S5A and S5B Fig), indicating that ISE6 cells possess molecular mechanisms to produce sRNAs that are different from those known in *Drosophila* or *C. elegans* (Fig 2A).

The sRNA species may regulate levels of the host ncRNA species. However, the level of the 300nt product of SRP RNA showed no clear difference between control and any of the knockdown samples (S5B Fig, second panel from the bottom). The effects of sRNAs on their host transcripts remain unclear.

## Chemical structures of tick sRNAs

The chemical structures of 5'- and 3'-terminal nucleotides of sRNAs often reflect their biogenesis mechanisms because RNA processing and modifying enzymes leave characteristic functional groups at these ends [34,35]. In general, AGO-bound sRNAs possess 5'-monophosphate groups that are recognized by the 5' binding pocket of the MID domain [36], with a notable exception of nematode secondary siRNAs which possess 5'-triphosphate groups [27]. piRNAs, fly siRNAs and plant miRNAs have 2'-O-methyl groups at their 3' nucleotides, whereas animal miRNAs carry hydroxyl groups at the equivalent position [37]. The 2' modification status at the 3'-nucleotide could be analyzed by oxidizing RNA samples with a periodate, as the presence of vicinal free 2'-, 3'-OH species makes the RNA molecule amenable to oxidization and resulting oxidized RNA molecules lack a 3'-OH group that is required for the 3' linker ligation for sRNA library construction [38,39]. Although piRNA species were efficiently enriched in our oxidized sRNA library (S4E Fig), sRNAs from RNAP III-transcribed genes were depleted, indicating that the latter had free 2'-, 3'-OH groups (Fig 2C). To further support this conclusion, we verified the results by Northern blotting. ß-elimination of the 3' nucleotides occurs when oxidized RNA species are incubated in an alkaline solution, resulting in faster migration ß-eliminated RNA species on the denaturing gel [40]. After ß-elimination, piRNAs remained at the same size, while miRNAs migrated more rapidly, consistent with the previously known 3' structures of their counterparts in other animals (Fig 2D). We observed faster migration of sRNAs from RNAP III-transcribed genes after ß-elimination, confirming the conclusion that they had 2'-OH species at the 3'-nucleotide. Although this was different from the known structure of the fly siRNA, recent reports also showed that TE-derived sRNAs in arachnids had free 2'-OH at their 3'-ends [8,10,11,26].

We also analyzed the 5' chemical structures of the sRNAs. The standard sRNA cloning protocol is selective for 5'-mono-phosphorylated species by taking advantage of the substrate

specificity of the RNA ligase [41]. The efficient inclusion of the sRNAs derived from RNAP III-transcribed genes in our libraries suggested that they harbored monophosphate groups at their 5' ends. To confirm this, we prepared an sRNA library after removing sRNA species with 5'-monophosphate groups by Terminator exonuclease [42] followed by dephosphorylation and re-phosphorylation of the 5' ends using T4 polynucleotide kinase, allowing the resulting libraries to enrich 5' di- or tri-phosphorylated sRNAs (Fig 2C, bottom). sRNAs from RNAP III-transcribed genes were not enriched in the 5' mono-P-depleted library when compared with the regular 5'-mono-P-enriched library, supporting the hypothesis that these sRNAs were 5'-mono-phosphorylated.

Taken together, we concluded that the novel sRNA species from RNAP III transcribed genes carried a 5'-mono-phosphorylated group and were not modified at the 2'-position of the 3'-nucleotide.

## Evolutionary conservation of sRNA production from RNAP III-transcribed genes

If RdRP-dependent sRNAs play important biological roles, one would expect the production of similar sRNA species to be conserved in evolutionarily distant tick species. We reanalyzed sRNA libraries from the Asian longhorned tick (*H. longicornis*) [43–45]. Phylogenetic analysis suggested that *H. longicornis* and *Ixodes* species shared the last common ancestor ~200 million years ago [46]. We identified sequences homologous to the RNase P, RNase MRP and SRP RNA genes in the *H. longicornis* genome, and found that sRNAs were mapped to both strands of these loci (Figs 2E and S6A). Importantly, they showed peaks at 22nt on both strands, suggesting that they were produced by specific processing machineries (Figs 2F and S6B).

Therefore, the production of 22nt species from RNAP III transcribed genes was broadly conserved in ticks. Furthermore, the presence of similar sRNA species in libraries made from tick animals and saliva suggested that the sRNA production was not restricted in cultured cell lines. The deep conservation of sRNA production from RNAP III transcribed genes suggests the biological importance of this sRNA class.

## Various sRNAs are produced from select coding genes

Although the fraction of sRNAs that mapped to coding exons was small (Fig 2A and S2 Table), the production of sRNAs from both strands suggested the involvement of RdRPs. We analyzed sRNA reads mapping to individual annotated protein-coding genes and collected genes that produced sRNA reads at >35RPM on average in the knockdown libraries (Fig 3A and S1 Data). Most of the loci showed a strong reduction (>40%) in at least one library compared to the GFP-KD control (28 out of 39 loci, Fig 2B and S1 Data). sRNAs from some loci were reduced in more than one sample with frequent overlaps between the RdRP3-Ago-16 and Aub-AGO3 combinations (Fig 3A). On the other hand, no locus showed reductions both in RdRP1- and RdRP3-knockdown libraries. These results suggested that certain combinations of factors formed sRNA processing pathways and the two RdRPs belonged to different pathways. The size distributions of sRNAs from individual loci roughly corresponded to their processing dependencies, where 22nt and 25-29nt peaks tended to be RdRP- and PIWI-dependent, respectively (Fig 3B, lower panels, S1 Data).

ISCI012234, which encodes a homolog of histone H1, produced the highest number of sRNAs and the sRNAs were dependent on RdRP3 and Ago-16 (Fig 3B). As mature histone mRNAs are generally not poly-adenylated and polyA tails inhibit the antisense production by plant RDR6 [48,49], an interesting possibility was that the polyA status might correlate with the sRNA production efficiency from the mRNA. We identified mRNAs enriched in rRNA-

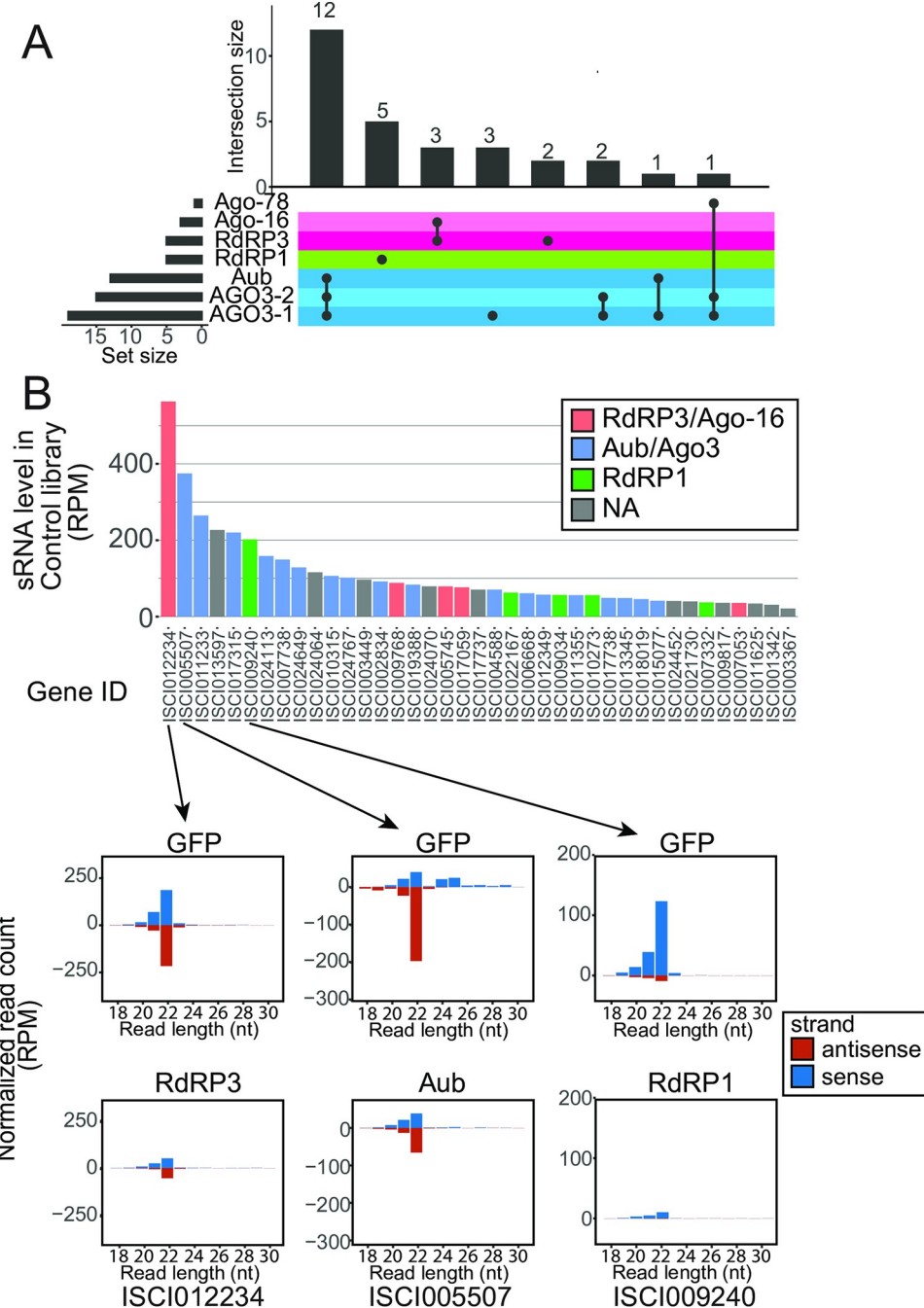

**Fig 3. sRNAs from coding regions.** (A) UpSet plot [47] of sRNA factor dependencies for sRNA-producing coding genes. The genes that had >35 RPM on average in the ten sRNA libraries were considered. If the sRNA reads from the locus were reduced by >40% in the knockdown library, the sRNA was judged as "dependent" on the factor that was knocked down. The number of dependent genes in each group is shown. Note that many genes were dependent on the Aub-AGO3 (blue) and RdRP3-Ago-16 (pink) combinations. (B) Read counts from protein-coding sequences in the control GFP-KD library are shown in the bar chart. sRNA reads derived from the annotated coding regions that had >35 RPM on average in the ten knockdown sRNA libraries are shown. If sRNA reads were reduced by >40% upon knockdown of one of the indicated sRNA factors, the gene was judged dependent on the factors. sRNA read size distributions of a representative locus in each group are shown in the bar charts below. If sRNA reads were reduced by >40% by knocking down any of the indicated sRNA factors in the group, bars are colored according to the following color-code (Blue: Aub-AGO3-1, or AGO3-2; Pink: RdRP3 or Ago-16; Green: RdRP1 or Ago-30).

depleted RNAseq libraries when compared with polyA-enriched RNAseq libraries [50] (S7A Fig, S3 Table) and tested whether the mRNAs produced sRNAs more efficiently than the other mRNAs. Indeed, we observed a slight but statistically significant negative correlation between the polyA status and the sRNA production (S7B Fig). Two other histone-like genes produced sRNAs at relatively high levels (S7A and S7C Fig). Many mRNAs were relatively strongly enriched in the rRNA-depleted RNAseq libraries without producing many sRNAs, suggesting that other factors also affect the sRNA production (S7C Fig). The production of sRNAs from PolyA (-) group was significantly higher than that from the polyA (+) group, supporting the hypothesis that polyA tails inhibit the production of RdRP-dependent sRNAs (S7D Fig).

Because ISCI012234 produced the highest sRNA read counts, we tested whether RdRP3-dependent sRNAs affect its expression. However, the expression level of this mRNA showed no consistent change (see below and S3 Table). Therefore, molecular functions of RdRP3-dependent sRNAs remained unknown. Global analysis of sRNAs from protein-coding genes revealed that multiple biogenesis mechanisms were involved in the production of coding gene-derived sRNAs. The fact that each RdRP was involved in the production of sRNAs from a small number of loci suggested that the RdRPs selectively recognize their substrates for sRNA production.

## sRNAs produced from repeats

A common role for metazoan RNAi/piRNA pathways is silencing of TEs [27]. To study TE-derived sRNAs, repetitive sequences were identified by the RepeatModeler2/RepeatMasker pipeline [51] and a genome-wide annotation of the repetitive sequences was obtained (Materials and Methods). To test which of the sRNA factors might be working together within the same sRNA biogenesis pathways, we counted the numbers of TEs whose sRNAs were commonly reduced in multiple knockdown libraries (Fig 4A). We used 67 repeats that produced abundant sRNA reads (>800 rpm on average in the knockdown libraries) for this analysis. As expected, the repetitive sequences often produced sRNAs that were reduced (<60%) upon knockdown of the PIWI genes (30 out of the 67 TEs examined), and many of these showed reduced levels of sRNAs in all of the three PIWI-family knockdown libraries (dsAub, dsAGO3-1 and dsAGO3-2, 13 out of the 30 TEs producing PIWI-family dependent sRNAs). Large overlaps were seen with Ago-16-RdRP3 and Ago-30-RdRP1 combinations, suggesting that the AGOs and RdRPs might work together to produce repeat associated-sRNAs. All these results again suggested that these groups of processing factors represented sRNA production pathways that largely independently operate to produce their own classes of sRNAs.

When we analyzed sRNA changes for individual repeat families, we found very similar sets of repeat families showed reductions in the piRNA abundance in the two AGO3 knockdown samples, whereas the Aub knockdown library showed a distinct response (S3F Fig). This result again supported the notion that the two dsRNA fragments against AGO3 indeed interfered with the expression of the single AGO3 gene.

To gain further insights, we analyzed individual repeat families (Fig 4B). We noticed that repeat families producing larger numbers of sRNAs tended to be RdRP-dependent whereas families that produced PIWI-dependent sRNAs tended to produce fewer sRNAs (Fig 4B, upper). When their sRNA sizes were analyzed, families that produced RdRP1- or RdRP3-dependent sRNAs showed clear 22nt peaks and families that produced PIWI-dependent sRNAs showed peaks at ~25-28nt (Fig 4B, bottom). The peaks at the expected sizes of their corresponding classes were strongly reduced by knockdown of the RdRP or PIWI protein, confirming the specific roles of these machineries in producing the respective classes of sRNAs. We verified biogenesis mechanisms of the TE-derived piRNA from family-423 (S5B Fig, second

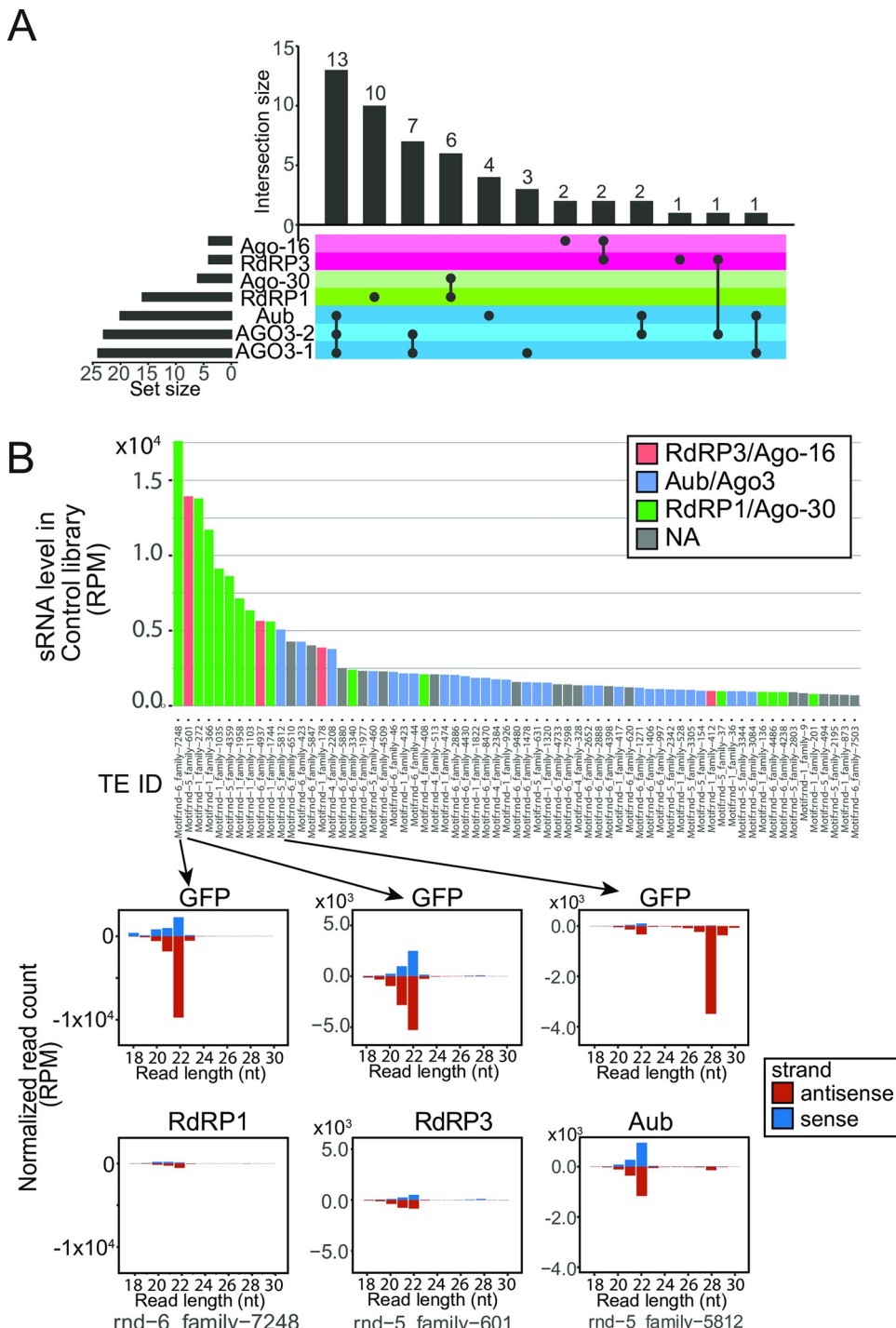

**Fig 4. sRNAs from repeats.** (A) UpSet plot [47] of sRNA factor dependencies for sRNA-producing repeats. The repeats that had >800 RPM on average in the ten sRNA libraries were considered. If the sRNA reads from the locus were reduced by >40% in the knockdown library, the sRNA was judged as "dependent" on the factor that was knocked down. The number of dependent genes in each group is shown. Note that many genes were dependent on the Aub-AGO3 (blue), RdRP3-Ago-16 (pink) and RdRP1-Ago-30 (green) combinations. (B) Read counts of repeat-associated sRNAs in the control GFP-KD library are shown in the bar chart. Repeats that had >800 RPM on average in the ten knockdown sRNA libraries are shown. If sRNA reads were reduced by >40% by knocking down any of the indicated sRNA factors in the group, bars are colored according to the following color-code (Blue: Aub-AGO3-1, or AGO3-2; Pink: RdRP3 or Ago-16; Green: RdRP1 or Ago-30). sRNA read size distributions of a representative repeat in each group are shown in the bar charts below.

panel). The ~27nt band recognized by the probe was strongly reduced by Aub or AGO3 knockdown, as expected from the sRNAseq results (S1 Data). In addition, we noticed that there was a less abundant species at ~22nt, whose expression was reduced in RdRP1-knock-down cells (S5B Fig). This indicated that some repeats produce both 22nt sRNAs and piRNAs. Furthermore, we occasionally observed repeats whose 22nt peaks were reduced (e.g. rnd-1_family-1111TE, S1 Data) or increased (e.g. rnd−5_family−5812, Fig 4B and S1 Data) upon Aub knockdown in addition to the reduction of the 25-28nt piRNA peaks. Therefore, interactions between these pathways should not be ruled out.

piRNAs are known to be amplified by a mechanism called the ping-pong cycle [31]. As some repeats produced piRNA-sized reads from both strands (S8A Fig), we wondered if there was a detectable signal for the amplification. piRNAs amplified by the ping-pong cycle exhibit characteristic 10-nt overlaps between sense and antisense reads, as a result of slicing of taget RNA by the PIWI protein, which cleaves the target RNA strand at the site complementary to the 10th and 11th nucleotides of the guide piRNA [52,53]. We analyzed the overlaps and detected significant 10nt-overlaps in the repeat-derived piRNA population (S8B Fig), indicating that ISE6 cells not only express two PIWI-clade Agos, but also they have an active ping-pong amplification mechanism.

To test if repeat-associated sRNAs regulate the expression of repeats, the levels of transcripts from repeats were analyzed after knocking down Ago-16, RdRP1 or RdRP3 (S4 Table). To our surprise, very few repeats were misregulated. The most significantly misregulated repeat in the Ago-16 knockdown libraries was rnd-6_family-4937, which was also most significantly misregulated in the RdRP3 knockdown libraries. As this repeat produced the second highest number of RdRP3-Ago-16 dependent sRNAs (Fig 4B), these results suggest that the RdRP3-Ago-16 axis may silence repeats.

## Roles for sRNA factors in gene regulation

To clarify whether the new sRNA pathways described here had roles in gene regulation, we analyzed the total RNAseq data of ISE6 cells after knocking down Ago-16, RdRP1 or RdRP3 (Fig 5).

Upon knockdown of these genes, we detected 47–84 genes to be differentially expressed compared to the control GFP KD sample (adjusted p-value <0.05, Fig 5A–5C and S3 Table). GO-term analysis revealed enrichment of the biological process categories related to RNAi and response to dsRNAs upon knockdown of Ago-16 or RdRP3 (Figs 5D and S5F, S1 File). In particular, Dicer homologs were upregulated in both libraries, while AGO homologs were up- and down-regulated in RdRP3 and Ago-16 knockdown libraries, respectively. Although the possibility of off-target effects of the introduced Ago-16 dsRNAs on their homologs could not be excluded, these results strongly suggested auto-regulation of the genes in sRNA-related pathways. Upon RdRP1 knockdown, stress-response-related genes were often down-regulated (S1 File).

Misregulation of the extracellular signal-regulated kinase (ERK) protein (ISCI005428, here-after Dsor1 after the fly gene name Dsor1) upon RdRP1-knockdown caught our attention because it was most strongly upregulated in this dataset (Fig 5B highlighted by blue). The annotation of *Ixodes* genes was incomplete and gene models generally lacked UTRs. We noticed that there was a strong peak of RdRP1-dependent sRNAs in the downstream region of the Dsor1 CDS (Fig 6A). The total RNAseq data showed continuous signals for ~4kb after the Dsor1 coding region, suggesting that the signal represented the 3' UTR of Dsor1 (Fig 6A). Consistent with this idea, RT-PCR using primers that bind the 3' end of the CDS and the 3' end of the putative 3' UTR yielded products having the correct sequence in a reverse-

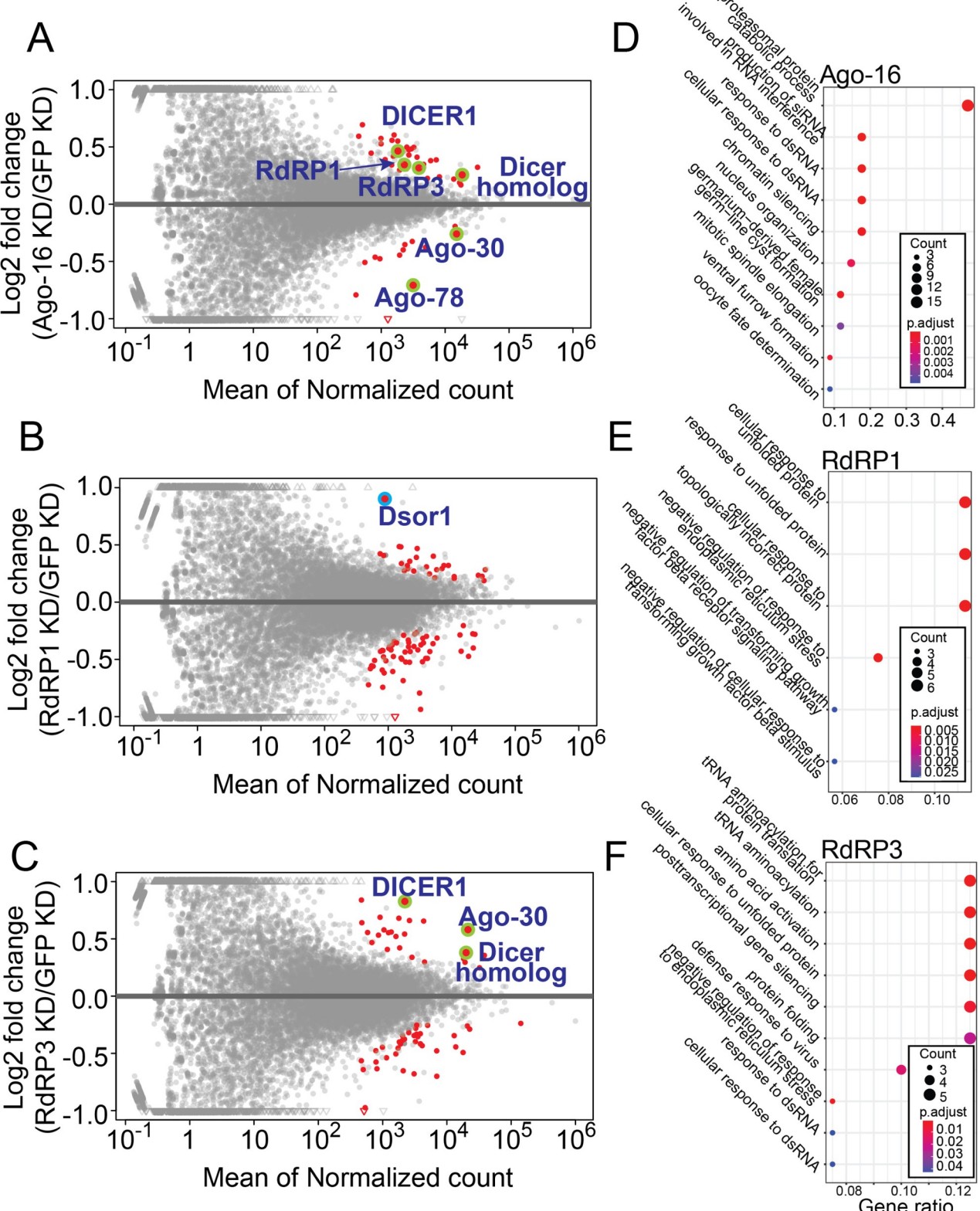

**Fig 5. Roles for the RdRP-dependent pathways in gene regulation.** (A-C) MA-plots of pairwise comparisons between the control (dsGFP) and the knockdown sample (n = 3). Red dots indicate genes differentially expressed in the knockdown samples (adjusted p <0.05) and RNAi-related gene names are indicated if they are differentially expressed and highlighted by green circles. The point representing Dsor1 was highlighted by a blue circle.

(D-F) GO-term analysis of misregulated genes revealed in the above analysis. Results of biological process analysis are shown. The size and color represent the number of genes in the GO category and the significance of enrichment, respectively, as indicated in the legend. Results of molecular function and cellular component groups analyses are shown in S1 File.

transcription-dependent manner (S9A and S9B Fig). The region where a large number of sRNAs were mapped corresponded to the rnd-1_family-272 sequence in our repeat annotation (S1 Table), which showed similarity to LTR/Gypsy family transposons (S1 Table). Therefore, the sRNAs targeting Dsor1 might be produced from other copies of this TE and act in trans.

The reciprocal changes in the targeting sRNAs and the target mRNA suggested direct regulation (Fig 6B and 6C). We first verified that Dsor1 was upregulated in RdRP1 knockdown cells by qPCR (Fig 6C, qPCR panel). A statistically significant increase in the Dsor1 level was also observed upon Ago-96 knockdown in addition to RdRP1 knockdown, suggesting that Ago-96 might also be involved in the regulation of Dsor1 (Fig 6C). To test if RdRP1 regulates Dsor1 through its 3'UTR, we cloned Dsor1 3'UTR after the firefly luciferase coding region of the pmirGLO/Fer-Luc2/Act-hRluc vector [54]. After depleting RdRP1 or RdRP3 in ISE6 cells, we transfected the Dsor1 3'UTR sensor plasmid and performed luciferase assays (Fig 6D). Upon knockdown of RdRP1, we detected ~3-fold upregulation (p = 0.002) of the sensor expression, whereas RdRP3 knockdown had no effect. These results demonstrated that RdRP1 regulates Dsor1 through the 3'UTR, presumably through the production of repeat associated sRNAs.

## Roles for sRNA factors in controlling viral RNA levels

Besides endogenous gene regulation, invertebrate RNAi pathways play roles in antiviral defense [55]. In ticks, a previous study demonstrated that RNAi factors including some AGOs were involved in controlling tick-borne viruses in tick cells and animals [28,45]. We were interested in testing if RdRPs control viral transcript levels in ISE6 cells.

In a recent study, a set of persistently infecting viruses in the ISE6 culture were identified by next-generation sequencing of putative viral particles [56]. We profiled sRNAs derived from the viral sequences in the knockdown sRNAseq libraries. In the control library, sRNAs mapping to the viral genomes were abundantly present (Fig 7A). The reads were distributed across the entire genomes with no strong enrichment in particular regions. The size distribution of the mapped reads showed a strong peak at 22nt without a recognizable peak at the piRNA size (Fig 7B). This was consistent with the vsiRNA seen in TBEV-infected ticks and cells [28,45].

For more comprehensive analysis, we de novo assembled viral contigs using the small RNA data from the GFP knockdown control library according to a previously described approach [57,58], and remapped sRNA reads in the GFP knockdown control library to the assembled viral contigs. This attempt yielded 54 contigs whose sequences show similarities to known eukaryotic viruses, including contigs corresponding to fragments of all of the five viral sequences used above. Even with this approach, no clear piRNA reads were observed, and no significant ping-pong signature was detected (S10A and S10B Fig, size distribution plots for individual viral contigs are in S1 Data). However, we note that these results do not exclude the possibility that ticks produce piRNAs against some viruses or under some conditions.

We sought to determine if vsiRNAs were dependent on any of the AGOs or RdRPs. Interestingly, vsiRNAs were still produced upon knockdown of any of the RNAi factors (Fig 7C, upper panel). This suggested that these sRNA factors were dispensable for vsiRNA production although functional redundancy between the factors might hinder the detection of the effects. We tested if the knockdown of these factors had any effects on the abundance of viral transcripts (Fig 7C lower panel). In our total RNAseq libraries, upregulation of some of the viruses

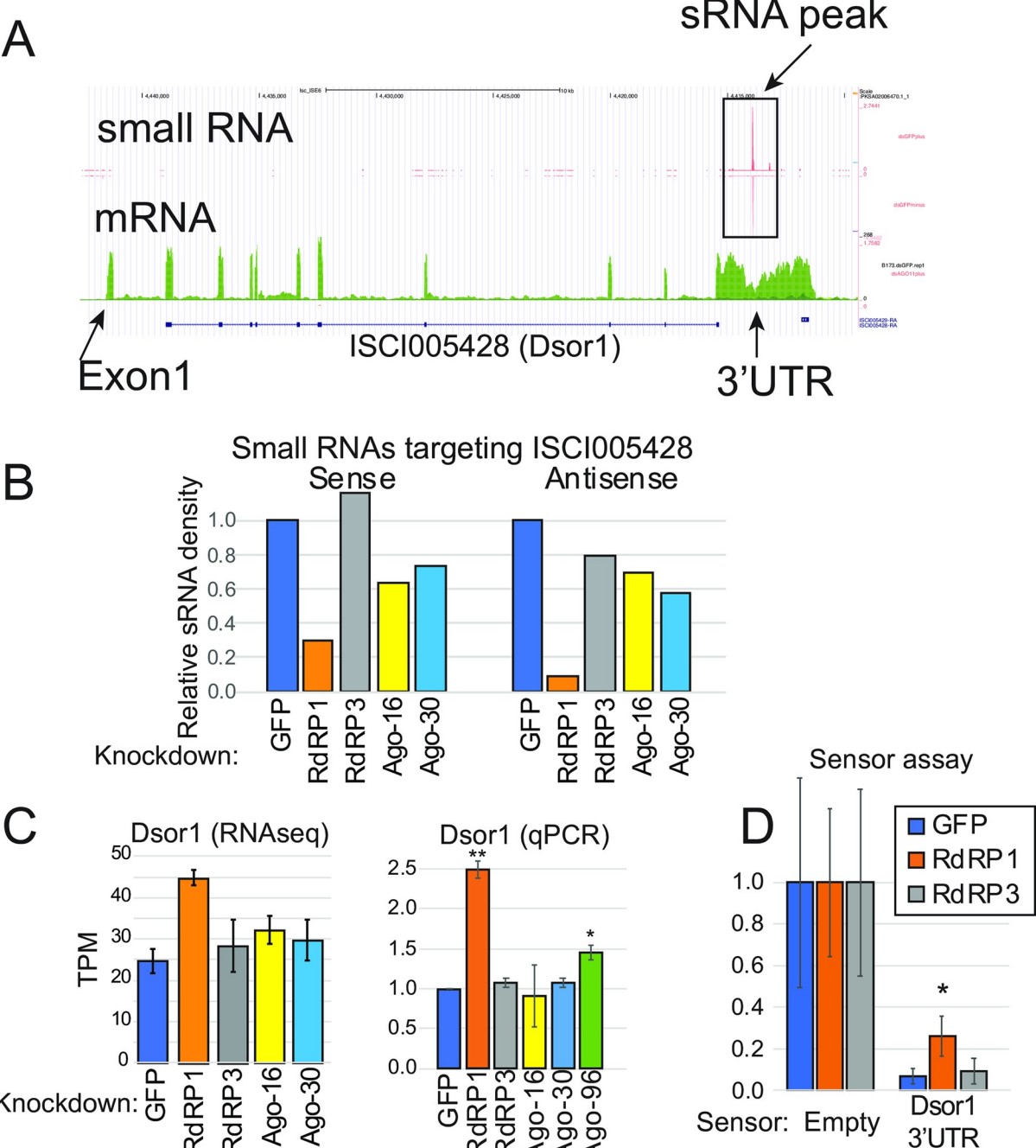

**Fig 6. Dsor1 is a target of the RdRP1-dependent sRNA pathway.** (A) Sreenshot of UCSC genome browser of the locus of the RdRP1-regulated gene Dsor1 (ISCI005428). The original gene structure annotation in the Vectorbase (ISCI005428) lacks the UTRs but reads corresponding to the extending transcript beyond the coding region are visible on the RNAseq mapping data. Within its 3'UTR, there is a high peak of sRNAs ("sRNA peak"). The Y-axis of the mRNA-seq and sRNA-seq mapping tracks means the normalized read counts. (B) sRNA reads mapping to the sense or antisense strands of Dsor1 locus was quantified. A dramatic reduction of sRNAs mapping to both sense and antisense strands was seen upon RdRP1 knockdown. (C) Expression levels of Dsor1 mRNA as analyzed by the salmon pipeline. The averages and standard deviations of TPM values are shown (n = 3). The results were verified by qPCR (Right panel, n = 4). (D) The Dsor1 3'UTR was cloned in the pmirGLO/Fer-Luc2/Act-hRluc vector [54], and ISE6 cells that were soaked with the indicated dsRNA were transfected with the sensor plasmid. The ratio between firefly luc and Rluc was normalized to that of the empty sensor, and the means and standard deviations are shown in the chart (n = 7). The experiments were performed twice on different days and the results from the two experiments were combined. Student's t-test was used to calculate p-values.

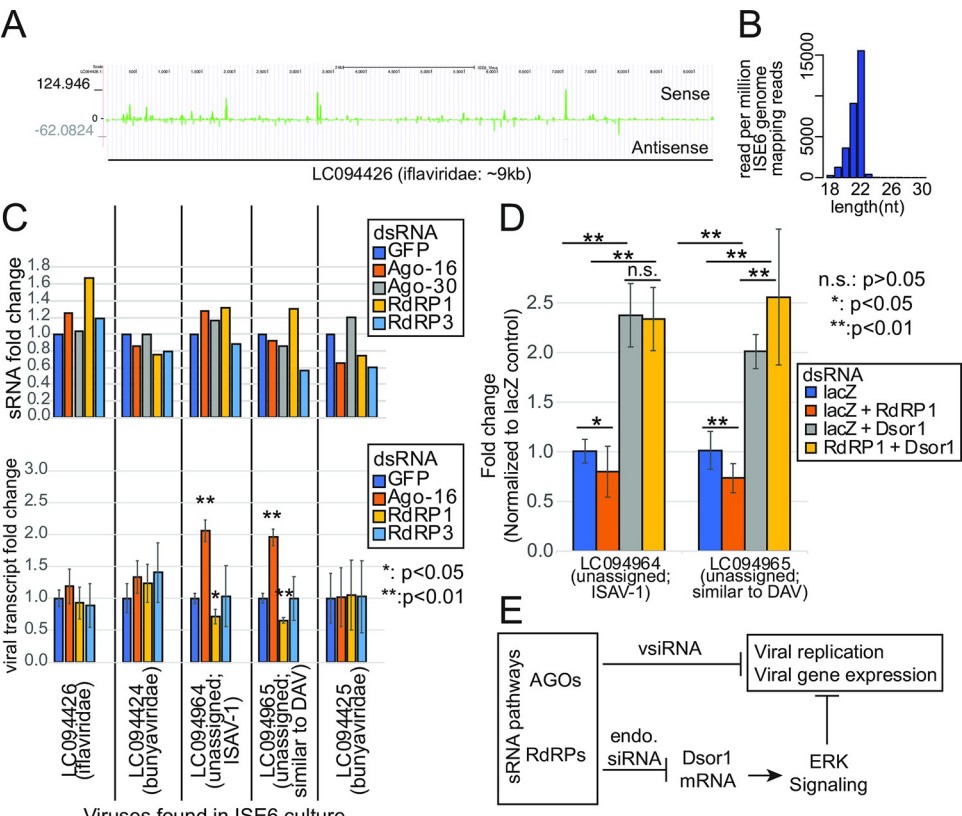

**Fig 7. sRNAs from persistently infecting viruses.** (A) sRNA reads from non-treated ISE6 cells were mapped to the 5 viral genomes that were known to be present in ISE6 cultures. As a representative virus, LC094426 (iflaviridae) is shown. Read density was calculated at each base and normalized for the number of reads mapping to the ISE6 genome. (B) Size distribution of viral sRNAs. Viral sRNAs of each length were counted and normalized to the number of reads mapping to the ISE6 genome (reads per million genome mapping reads). The result of the non-treated control cells is shown. (C) Changes in the sRNA (Upper, n = 1) and the viral transcript (Lower, n = 3) abundances in ISE6 cells after knocking down AGO or RdRP. sRNA libraries and total RNAseq libraries were analyzed to quantify the abundances of viral sRNAs and viral transcripts in the indicated libraries. Fold change values were calculated based on reads per million genome mapping reads (for sRNAs) and TPM (for viral transcripts). The averages and the standard deviations are shown. The p-values were calculated by comparing each group with the control (dsGFP) group and the asterisks indicate the significance (*: p<0.05, **: p<0.01; n = 3, t-test). (D) Changes in the viral transcript level upon RdRP1-Dsor1 double knockdown. Cells treated with the indicated dsRNAs were used. The amount of dsRNA was adjusted by adding lacZ control dsRNA so that each sample is treated by the same amount of dsRNA. qPCR primers detecting the indicated viruses were used and the values were normalized by actin and expressed as fold change relative to the level in the lacZ dsRNA control sample. T-tests were performed in all combinations and the number of asterisks indicate the significance (n.s.: p>0.05, *: p<0.05, **: p<0.01; n = 12). (E) Working hypothesis. The tick has various sRNA pathways, some of which control viral transcripts by vsiRNAs and some others control antiviral response by controlling Dsor1 expression by RdRP1-dependent endogenous sRNAs.

upon knockdown of Ago-16 was observed (Fig 7C, lower panel), consistent with the previous study using TBEV/LGTV [28]. Furthermore, we observed unexpected downregulation of some viruses upon knockdown of RdRP1 (Fig 7C, lower panel). While we observed no reduction of vsiRNA in RdRP1 knockdown samples, these results suggested that RdRP1 might have a role in maintaining the levels of viruses independently from vsiRNA production.

We hypothesized that the reduced viral gene expression in RdRP1-knockdown cells was due to an enhanced immune response by upregulation of Dsor1. To test this, we knocked down RdRP1 and Dsor1 simultaneously (Fig 7D). As expected, Dsor1 knockdown caused upregulation of viral transcripts, and RdRP1 knockdown caused a decrease of viral transcripts,

consistent with our RNAseq data. When both RdRP1 and Dsor1 were knocked down together, no significant downregulation was observed compared with Dsor1 single knockdown. These results suggested that RdRP1 downregulated viral transcripts via upregulation of Dsor1.

In summary, our results showed that the components of the sRNA pathway play roles in the regulation of mRNA expression primarily to regulate genes related to sRNA pathways. Furthermore, RdRPs appeared to be involved in controlling viral transcripts in a vsiRNA-independent manner. While the biological significance of these mechanisms in normal gene regulation and virus-tick interactions needs to be studied in the future, this study unveiled novel and unexpected regulatory mechanisms involving tick-specific sRNA factors (Fig 7E).

## Discussion

In contrast to the established roles of RdRPs in plants, fungi and worms, their roles remain unclear in other animals. Although RdRP genes were found in many arthropods, their roles in sRNA production were not experimentally demonstrated mainly due to the lack of suitable experimental systems [8–11]. Analysis of sRNA chemical structures of sRNAs from various animals possessing RdRP genes did not find evidence for the production of sRNAs containing terminal structures similar to those of RdRP-dependent sRNAs in worms [26]. Based on these results, the importance of RdRPs in animals outside of Nematoda remained controversial.

In the present study, we demonstrated the presence of abundant classes of RdRP-dependent sRNAs in tick cells. Some of them are expressed as highly as the most abundant miRNA genes expressed in the cell line (Fig 2A), implying that they played important biological roles. The conservation of the catalytic site in the tick RdRPs suggested they are active enzymes potentially producing antisense RNA species [59,60].

A large fraction of RdRP-dependent sRNAs was derived from RNAP III-transcribed genes (Fig 2A), pointing to a potential functional link. Transcription by RNAP III is terminated by the presence of a stretch of 5 or more Ts on the non-template strand [61] and the presence of short U-tails is a signal for clearance by the quality control mechanisms [62–64]. Therefore, these U-tails may also act as a signal for RdRP1 to produce their antisense RNAs in ticks. In fact, in *C. elegans*, 3'-oligouridylation acts as a signal for RdRP-dependent sRNA production in artificial RNAi or silencing of rRNA transcription upon erroneous pre-rRNA production [65,66]. In addition, histone mRNAs are oligourydilated before they are subjected to degradation [49]. Therefore, oligo-U tails may be a signal for RdRP recognition.

However, the distance between the positions of abundant tick sRNAs and the 3'-ends of RNAP III transcripts showed no obvious trends in contrast to the expectation that the antisense RNA production may be initiated at a certain distance from the U-tails [65]. Additional evidence to support this hypothesis is lacking. Alternatively, RdRP1 might physically interact with RNAP III during transcription, similarly to how RDR2 in *Arabidopsis* recognizes RNAP IV products to produce their antisense strands [60]. The functional links between RdRP1 and the RNAP III machineries remain unclear, and this deserves further investigation.

What might be the roles for the tick endogenous sRNA pathways? The production of antisense sRNAs from RNAP III-dependent genes appears to be conserved in the two tick species, *I. scapularis* and *H. longicornis*, suggesting that this pathway has a conserved role (Fig 2E and 2F). However, we observed no discernible effects of RdRP1-KD on the expression levels of the RNAP III product SRP RNA, suggesting that the sRNAs may play roles independently of gene regulation in cis (S5B Fig). In fission yeast, genes transcribed by RNAP III are involved in the compartmentalization of the genome in the nucleus by defining boundaries between regions with distinct chromosomal states [67,68]. It is interesting to speculate that the tick sRNA pathway might also contribute to the higher order organization of the genome in the tick nucleus.

On the other hand, as demonstrated with Dsor1, tick sRNAs can also down-regulate mRNAs containing sRNA targets at least in some cases (Fig 6).

The function of RdRP3-dependent pathway is even more mysterious. RdRP3 knockdown caused misregulation of 63 genes (S3 Table), many of which were also misregulated in Ago-16 knockdown (Fig 5A and 5C). This was consistent with the overlapping dependencies of sRNAs on these factors (Figs 3 and 4). The misregulated genes included many sRNA-related factors including Dicer homologs. Autoregulatory mechanisms have also been described in worms, suggesting that this is a common phenomenon [69]. However, as these loci do not produce abundant RdRP3-dependent sRNA species, how the RdRP3-Ago-16 axis controls gene expression remains unknown. It is important to identify direct targets to understand their molecular functions. Studying gene regulation using multiple approaches such as proteomics and chromatin structure analysis will be important. Characterization of biochemical properties of the AGOs may also provide clues to the molecular functions for the novel sRNA species. In addition to molecular analysis, biological roles for RdRP-dependent sRNAs need to be investigated especially in the in vivo context.

One of the main roles for invertebrate RNAi pathways is the antiviral response. In animals including worms, the roles for RdRPs in anti-viral defense mechanisms remain to be studied. In plants, an *Arabidopsis* mutant of the RdRP gene RDR6 exhibited normal antiviral responses, suggesting that RDR6 is dispensable [23,70]. On the other hand, in tobacco, knockdown of the RDR6 gene caused higher susceptibility to viruses especially at high temperatures [71]. Viruses often encode proteins to suppress hosts' RNAi mechanisms (Viral Suppressors of RNA silencing or VSRs), and complex interactions between VSRs and RNAi factors often complicate the interpretation of experimental data [72]. Therefore, the contribution of VSRs needs to be taken into consideration to understand interactions between viruses and host or vector cells. The knowledge regarding the RdRP-dependent sRNA pathways obtained here will help us untangle the complex interactions at the molecular level.

Various signaling pathways, including the ERK pathway, play roles in antiviral immunity [73]. It is interesting that the integral ERK pathway factor, Dsor1 is upregulated upon RdRP1 knockdown in tick cells, suggesting that RdRP1 might negatively regulate the immune response (Fig 6). In addition, some of the persistently infecting viruses were reduced upon RdRP1 knockdown whereas knockdown of Ago-16 tended to have the opposite effect (Fig 7). Therefore, our results suggested complex interactions between the host sRNA pathways and the virus rather than simple antiviral roles of sRNA factors. This is at least in part mediated by Dsor1 as RdRP1 knockdown did not decrease viral transcripts when Dsor1 was also knocked down. The direct and indirect roles of tick RdRPs in the life-cycle of tick-bone viruses should be studied in the future.

Tick cell lines have been used as an in vitro model to study interactions between tick-borne viruses and vectors [74]. Based upon the previously established tools, we provide a genomics platform to study the ticks' sRNA pathways and viruses by annotating endogenous sRNAs and their biogenesis pathways. For convenience, we made the genomics resources available to the research community, including ISE6 and *H. longicornis* UCSC genome browser assembly tracks, RNAseq mapping data tracks and the sRNA size distribution charts. The UCSC assembly hubs with the RNAseq mapping tracks are available at https://data.cyverse.org/dav-anon/iplant/home/okamuralab/trackhub/Isc_ISE6/IscaI1_hub.txt and https://data.cyverse.org/dav-anon/iplant/home/okamuralab/trackhub/ucscgb_haeL2018/hubHaeL2018.txt.Together with the initial characterization of the sRNA pathways presented in this study, these resources will facilitate studies related to gene regulation and virus-vector interactions that are mediated by sRNAs. Furthermore, as RNAi technologies depend on sRNA factors, detailed understanding

through the characterization of the sRNA factors may pave the way for the development of RNAi-based pesticides.

## Materials and methods

### Tick cell culture and dsRNA transfection

*Ixodes scapularis* embryonic 6 (ISE6) cells were obtained from ATCC and cultured according to the published protocol at 34 degrees C [75]. The dsRNA transfection was performed using Effectene (QIAGEN). Cells were seeded at 1x10^6 /ml in 2ml fresh L-15B medium on a 6-well plate. 400ng dsRNA was diluted in 100ul Buffer EC and 3.2ul Enhancer was added. The mixture was incubated for 2-5min at room temperature. Then, 10ul Effectene was added and incubated for 5-10min at room temperature. The mixture was added to the culture and the cells were incubated for 7–10 days. After the first incubation, the dsRNA transfection procedure was repeated again and incubated for 7–10 days to ensure the maximal efficacy of RNAi.

### Plasmids and dsRNA production

cDNA was amplified using the total RNA of ISE6 cells as a template. Contaminating genomic DNA was removed by treating total RNA samples with the Turbo DNA-free kit (Ambion) and cDNA was synthesized using Superscript III (Invitrogen) according to the manufactures' instructions. The cDNA encoding Ago-16, Ago-30, RdRP1 and RdRP3 were amplified using primers listed in S5 Table, and clones were obtained by inserting the amplified cDNAs in the NotI-XbaI sites of pEGFP with a modified multiple-cloning-site sequence [76]. The sequences of the inserts were verified by sequencing.

For the preparation of templates for dsRNA synthesis, ~500bp fragments of sRNA factors were amplified from cDNA of ISE6 cells using primers listed in S1 Table using iProof High-Fidelity Master Mix (Bio-Rad) and the amplicons were treated with XhoI to insert them in the XhoI site of pLitmus28i (NEB). To obtain templates for in vitro transcription, LitmusA and LitmusB primers were used (S5 Table). 5ul of the PCR product was used in a 20ul in vitro transcription reaction using Megascript T7 kit (Ambion). dsRNA was purified by Phenol/Chloroform extraction followed by ethanol precipitation as described previously [76].

### RNA extraction and RNAseq library preparation

Total RNA was extracted from ISE6 cells using Trizol-LS (Invitrogen) according to the manufacturer's instructions. For sRNA libraries of ISE6 cells depleted of sRNA factors and their control samples, 1ug total RNA was used for library construction using the TruSeq Small RNA Library Preparation kit (Illumina). For 5'-tri-P libraries, the sRNA fraction (~15-35nt) was isolated by gel extraction and RNA species bearing 5'-OH were monophosphorylated by T4 polynucleotide kinase (NEB). This was followed by treatment by a terminator exonuclease (Epicentre), dephosphorylation by Calf Intestine Phosphatase (NEB) and re-phosphorylation by T4 polynucletide kinase (NEB). For the oxidized sRNA library, the sRNA fraction (~15-35nt) from 50 ug total RNA was isolated by gel extraction and treated with 25mM $NaIO_4$ dissolved in 60 mM Borax buffer and incubated for 30 minutes at room temperature in the dark. Both 5'-tir-P enriched and oxidized samples were subjected to a sRNA library construction by the TruSeq Small RNA Library Preparation kit (Illumina). The resulting libraries were sent to BGI (Hongkong) for sequencing on a Hiseq2000.

For total RNAseq analysis, three sets of knockdown experiments were performed independently, and total RNA samples extracted by Trizol-LS (Invitrogen) were sent to BGI (Hongkong) for ribosomal RNA depletion using Ribo-Zero Gold rRNA Removal Kit (Human/

Mouse/Rat) (Epicentre) followed by library construction using non-stranded (Replicate 1) or stranded (Replicates 2 and 3) TruSeq mRNA Library Prep Kit (Illumina). The resulting libraries were sequenced on Hiseq4000.

## Identification of repetitive sequences

Repetitive sequences in the ISE6 genome (IscaI1; https://metazoa.ensembl.org/Ixodes_scapularis_ise6/Info/Index) were identified using the RepeatModeler2 pipeline [51]. The reference TE sequences were downloaded from Repbase < https://www.girinst.org/server/RepBase/> and the RepBaseRepeatMaskerEdition-20181026.tar file was used for annotation.

## Analysis of sRNAseq data

sRNA libraries were analyzed following the previously established pipeline [77]. Adaptor sequences were removed by using fastx_clipper and collapsed by fastx_collapser and converted by fasta_formatter (The tools were downloaded from <http://hannonlab.cshl.edu/fastx_toolkit/commandline.html>). The fasta files were used for mapping using the ISE6 genome sequence (IscaI1) [78] using bowtie1.3.0 < http://bowtie-bio.sourceforge.net/index.shtml> without allowing any mismatches. For identification of sRNA sources, genome-mapping reads were mapped to the following reference sequences: 1. miRNAs (miRbase Release 22.1) [79], RNAP III transcripts (downloaded from RNAcentral) [80], rRNAs from NCBI and mRNAs (Vectorbase IscaI1.0) [78]. As 5S rRNA loci are separated from the rDNA repeats and transcribed by RNAP III, we added the 5S rRNA sequences to the "RNAP III" group. The reference sequences are summarized in S1 Table. VirusDetect [58] was used to assemble non-redundant contig sequences based on sRNA-seq data from the control library. The assembled contig sequences were utilized for sequence similarity searches using BlastX against the NCBI non-redundant databases with taxonomy specified to be Viruses. The assembled contigs that hit the databases were taken as references for sRNA mapping using bowtie. Based on the bowtie output, sRNA size distribution with 5'base composition was analyzed using a python script adapted from [11] and ping-pong overlap signatures were computed by a python script adapted from [81].

## Phylogenetic tree construction

Based on multiple sequence alignment with MUSCLE, we used ModelFinder [82] to determine the best-fit model and obtained branch supports with the ultrafast bootstrap [83] implemented in the IQ-TREE software [84]. The structural prediction was performed on the ColabFold [85,86].

## Analysis of total RNAseq data

Total RNAseq libraries were analyzed as described previously [77]. The adaptor sequences were trimmed using Cutadapt [87] with the default quality cutoff value. Gene expression was quantified by salmon [88] using the Vectorbase IscaI1.0 annotation or a fasta file containing the viral genome sequences persistently present in the ISE6 culture (S2 Table; Nakao et al., 2017). Genome mapping was performed by bowtie2 <http://bowtie-bio.sourceforge.net/bowtie2/index.shtml> with the default setting. Differential gene expression analysis was done using the DESeq2 package [89], with the cut-off of adjusted-p-value set to 0.05. GO-term enrichment analysis of misregulated genes was conducted using clusterProfiler [90], with GO terms obtained from eggNOG-mapper [91] and TRAPID [92]. The transcriptome was assembled using Trinity [93]. A blast database was constructed using the assembled cDNA

sequences, and homologs of RNAi factors were identified by tblastn https://blast.ncbi.nlm.nih.gov/Blast.cgi?CMD=Web&PAGE_TYPE=BlastDocs&DOC_TYPE=Download using the bait sequences (RRF-1, EGO-1 and RRF-3 from *C. elegans*; AGO1, AGO2, Aub, AGO3 and PIWI from *D. melanogaster*). MUSCLE (Figs 1A and S1) or CLUSTAL-O (S3 Fig) was used for alignment.

### Northern blotting

Northern blotting was done as described previously [76]. Briefly, 10ug total RNA samples were separated on a 15% Sequagel (National Diagnostics) and transferred onto a positively charged nylon membrane and hybridized using DNA or LNA probes. The probe sequences are listed in S5 Table.

### RT-qPCR

10μg total RNA isolated from ISE6 cells was subjected to DNase I treatment (Ambion) according to the manufacturer's instructions. Reverse transcription was performed using iScript Reverse Transcription Supermix (Bio-Rad). qPCR was performed using iTaq Universal SYBR Green Supermix (Bio-Rad). Primers used for qPCR are shown in S5 Table. Actin was used as a reference gene for normalization.

### Sensor assay

dsRNA transfected cells were seed 1x10^6 /ml in 500 μl L-15B medium on a 24-well plate. Then, sensor plasmids were transfected using Effectene transfection reagent (QIAGEN) according to the manufacturer's instructions. Cells were incubated 4–5 days. Luciferase activity is measured using Dual-Luciferase Reporter Assay System (Promega). We first calculated the ratio of fluc and R-luc readings from each well, and the ratio was further normalized to the mean ratio of the empty sensor wells on the same plate.

### Western blotting

Western blotting was performed as previously described [94]. S5 Table lists antibodies used in this study.

### Accession number

The small RNA library data produced for this study are deposited at NCBI SRA under GSE183810.

## Supporting information

**S1 Fig. Alignment of RdRP genes.** The sequences of RdRP homologs from Fission yeast (Spo), Neurospora (Ncr), Arabidopsis (Ath), worm (Cel) and tick (Isc) were aligned by MUSCLE <https://www.ebi.ac.uk/Tools/msa/muscle/> and the regions near the catalytically active site and the Rrf1-specific putative loop are shown. The accession numbers of the gene sequences can be found in S1 Table.
(TIF)

**S2 Fig. Alignment of AGO genes.** The sequences of AGO homologs from *Drosophila* (dAGOs), human (hAGOs) and tick (IscAGOs) were aligned by CLUSTAL Omega < https://www.ebi.ac.uk/Tools/msa/clustalo/> and the region containing the catalytic residues is shown. The conserved DEDH residues are highlighted in red. The accession numbers of the

gene sequences can be found in S1 Table.
(TIF)

**S3 Fig. Characterization of AGO/RdRP proteins.** (A) Verification of gene knockdown. RNA samples isolated from ISE6 cells transfected with the indicated dsRNAs were used for qPCR verification. The qPCR primers were designed to quantify the expression of the cognate gene but avoided the region corresponding to the introduced dsRNA sequence. The values were normalized by the Actin expression and further normalized to the value in the dsGFP control sample. Four technical replicates were made and the averages and the standard deviations are shown. (B) ISCI021408 and ISCI004800 correspond to two fragments of the IscAGO3 gene. There are two loci in the ISE6 assembly that match with ISCI021408 and ISCI004800 genes at high homologies (>97.8%) and the gene structures were conserved in the two loci, suggesting that these two loci may represent a very recent gene duplication or a genome assembly artifact. (C) qPCR verification of AGO3 knockdown against ISCI021408 (dsAGO3-1) and ISCI004800 (dsAGO3-2). Note that transfection of either dsRNA construct resulted in strong reduction of both ISCI021408 and ISCI004800 qPCR amplicons, indicating the dsRNA constructs reduce both genes. Based on the observations in (B), we consider the two loci as a single gene, IscAGO3. (D) Western blot analysis for the PIWI proteins detected by anti-symmetric dimethyl arginine antibody, SYM11. ISE6 cells were treated with the indicated dsRNAs and total protein was extracted and separated on an SDS-PAGE gel. The proteins were detected by SYM11, which recognizes symmetric dimethyl arginine that is a characteristic of PIWI proteins and some other RNA binding proteins [32]. The quantified values of the band at ~100kDa were normalized to the value of the Actin band and shown in the bar chart next to the Western blotting panel. Note the strong reduction of the 100kDa band upon Aub KD, suggesting that this band corresponded to the Aub protein. (E) HEK293T cells were transfected with the indicated construct and total protein samples were subjected to Western blot analysis using a GFP antibody. The bands were detected at the predicted sizes. (F) Scatter plots of repeat-derived 25-30nt sRNA abundance in Aub, AGO3-1 or AGO3-2 knockdown libraries. Fold change values were calculated for each repeat family in the three knockdown libraries with respect to the RPM value in the control GFP knockdown library. Repeat families with at least 16 RPM in the control library(25-30nt) and 3 RPM in the three knockdown libraries(25-30nt) were considered. The red line represents the overall trend and the formula of the trend line is shown in each panel. The size of dots represent the number of 25-30nt reads in the GFP knockdown control library mapped to the repeat family.
(TIF)

**S4 Fig. Compositions of sRNA libraries.** (A) Size distribution of sRNA reads. Genome-matching sRNA reads of each length in the control (dsGFP) library were counted and normalized by the number of reads of all lengths and expressed as RPM (Reads Per Million mapped reads). (B) Percent stacked-bar-chart showing the composition of sRNAs in the knockdown libraries. The fractions of sRNAs derived from miRNAs (blue), RNAP III transcribed genes (pink), rRNAs (orange), snoRNAs + mRNAs (yellow), repeats (green) and sRNAs that did not belong to any of the categories (light green) are shown. 18-30nt reads were used. Note that the same bars are shown in Fig 2A for dsGFP, dsAgo-78, dsRdRP1 and dsRdRP3. (C) Percent stacked-bar-chart showing the composition of sRNAs in the control (dsGFP) library. 22nt (left bar) and 25-30nt (right bar) reads were used. (D) Normalized read count (RPM) of sense and antisense (22nt reads, left; 25-30nt reads, right) that were derived from repeats in the indicated knockdown libraries are shown in blue and red, respectively. Negative values were given to antisense read counts. (E) UCSC Genome Browser screenshot of the TE family-423 region. A strong peak of a piRNA (family-423-A) was observed in the control library, which was

prepared with the standard protocol. This peak was even higher in the library prepared after oxidization of the total RNA, supporting that this RNA species have a 2'-O-me group similar to piRNAs in other organisms. These species were depleted when 5'-mono-phosphorylated species were removed, suggesting that this RNA species had a 5'-mono-phosphate group. (TIF)

**S5 Fig. Biogenesis of sRNAs from RNAP III-transcribed genes.** (A) UCSC screenshot at the SRP RNA locus shown as an example of a locus producing RdRP1-dependent sRNAs. The tracks show sRNA mapping densities in the indicated libraries from the sense (upward) and antisense (downward) strands of the locus with respect to the direction of the transcription. (B) Verification of sRNA library analysis by Northern blotting of representative sRNA species. 10ug total RNA from each of the indicated knockdown samples was loaded, and the membrane was incubated with the indicated probes. The probe sequences can be found in S5 Table. The >150nt (SRP RNA), ~50-60nt (pre-mir-8) and ~20-30nt (other panels) areas of the blots are cropped and shown. miRNA-candidate-1 was not in miRBase but identified by visual inspection of mapping data as well as RNA secondary structure prediction for its precursor structure. The full analysis for novel miRNA genes will be published elsewhere. The SYBR Green II staining result is shown as the loading control. Red arrowheads show the bands that showed a clear reduction in the knockdown lanes. (TIF)

**S6 Fig. Conservation of 22nt-sRNA production from RNAP III transcribed genes in *H. longicornis.*** (A) UCSC genome browser screenshot of the RNase P locus in the *H. longicornis* genome. sRNA mapping density is shown. The upper (purple) and lower (blue) signals represent sense and antisense reads. The sRNAseq data are from previous studies [43,44]. (B) Size distribution of sRNAs mapping to the representative RNAP III-dependent genes (RNase P, RNase MRP and SRP RNA) of *H. longicornis*. The upper left and lower right panels are reanalysis of data from [43] and [45], respectively. Other panels are using [44]. For both (A) and (B), the saliva data are also shown in Fig 2E and 2F. (TIF)

**S7 Fig. PolyA status of mRNA correlates with sRNA production rate.** (A) Identification of mRNAs enriched in rRNA-depleted RNAseq libraries. Log2 of the average TPM values from ISE6 cells in total RNAseq libraries (this study; transfected with dsGFP) or polyA-enriched RNAseq libraries (control samples)[50]. To determine the cut off value, the linear regression line of spliceosomal RNA datapoints (red) was used. The green and grey dots represent mRNAs that were judged as polyA(-) and polyA(+) coding RNAs, respectively. Because of the enrichment cut off, polyA(-) coding RNAs had at least log2(TPM+1)> 3.288885. Therefore, only polyA(+) coding RNAs with log2(TPM+1)> 3.288885 were used for analysis in (B) to account for gene expression levels. The histone homologs found in the polyA(-) group are highlighted by larger dots and their gene IDs are shown. (B) The distributions of sRNA production values of polyA(-) (Pink line) and polyA(+) (Green line) coding RNAs are shown in a CDF plot. The RPM normalized sRNA counts for polyA(-) and polyA(+) coding RNAs were used. The X-axis shows log2 of the RPM value +1 in the sRNA libraries using ISE6 cells transfected with dsGFP. The p-value was calculated by the Wilcoxon test. (C) The sRNA production value (log2 of RPM value +1) of each polyA (-) coding gene was plotted against the enrichment factor in the total RNAseq library (log2(total RNAseq TPM +1)–log2(polyA RNAseq +1)). There are many genes showing strong enrichment in the total RNAseq library while not producing many sRNAs. The histone homologs found in the polyA(-) group are highlighted by red circles and their gene IDs are shown. (D) RdRP-dependent sRNA production from polyA

(+) and (-) groups. The loci producing sRNAs that are RdRP1-dependent (green), RdRP3-dependent (blue) and dependent on both RdRPs (pink) are shown. The genes were grouped based on the analysis shown in Fig 3. Y-axis shows the sRNA level in the control dsGFP library (log2 of RPM+1). The p-value was calculated by the Wilcoxon test.
(TIF)

**S8 Fig. Evidence for Ping-pong amplification of repeat-derived sRNAs in ISE6.** (A) Heat map of sense/antisense 25-30nt sRNA levels from individual repeat families in the control library. Repeats with at least 16 RPM 25-30nt reads were used. (B) The 5'-5' overlap between sense and antisense repeat-derived 25-30nt sRNAs was analyzed and the numbers of read pairs with 1-25nt overlaps (left) and the significance of the overlap (Z-score: right) are shown. A significant signal was observed at 10nt (Z score> 2.58 corresponds to p<0.01).
(TIF)

**S9 Fig. RT-PCR verification of Dsor1 3'UTR.** (A) UCSC genome browser screenshot of the Dsor1 (ISCI005428) locus in the *I.scapularis* genome. sRNAseq read density (upper) and stranded total RNAseq read density from the control libraries are shown. The positions of RT-PCR primers used in (B) are shown. The forward primer was designed against the 3'-end of the coding region, and the reverse primers were designed against the putative 3'UTR sequence. (B) RT-PCR results. Using the primers illustrated in (A), RT-PCR was performed with (+) or without (-) adding reverse transcriptase in the RT reaction. Both primer pairs amplified the expected fragments in a reverse-transcription dependent manner, confirming the extended sequence represents the 3'UTR of Dsor1. The band amplified with pair B was cloned and sequenced, and used for the luciferase assays shown in Fig 6D.
(TIF)

**S10 Fig. No evidence for viral piRNAs against persistently infecting viruses in ISE6.** (A) Viral sequence contigs were assembled using ISE6 sRNA library data and reads were remapped to the contigs. Reads were grouped based on the 5' nucleotides and the size distribution is shown. The X- and Y-axes show the size and the raw read count, respectively. The inset shows an enlarged view of the 25-30nt window with a smaller Y-scale. (B) The 5'-5' overlaps between sense and antisense virus-derived 25-30nt sRNAs were analyzed and the number of read pairs with 1-25nt overlaps (left) and the significance of the overlap (Z-score: right) are shown. No significant signal was observed at 10nt, suggesting the lack of ping-pong amplification of viral sRNAs.
(TIF)

**S1 Table. General bioinformatics information.** Sheet1: Statistics of the sRNAseq libraries. Sheet2: Reference sequences used for classification of sRNA origins. Sheet3: ISCI IDs of genes with sequences that matched with the dsRNA sequences used for the knockdown. Sheet4: Trinity contigs and Gene IDs of the AGO/RdRP genes analyzed in this study. Sheet5: References for amino acid sequences of AGO/RdRPs used for the phylogenetic analysis. Sheet6: Repeat families identified by RepeatModeler2/RepeatMasker. Sheet7: Statistics of the total RNAseq libraries. Sheet8: List of the *H. longicornis* sRNA libraries analyzed in this study.
(XLSX)

**S2 Table. Summary of sRNA analysis.** Sheets1-10: Related to Figs 2A and S4. sRNA read counts for each category are shown. Each sheet reports the numbers in each of the knockdown libraries. The sum of sense and antisense reads (C18-C30), the number of sense reads (S18-S30) and the number of antisense reads (AS18-AS30) that matched the reference sequence in the category in the header row are reported for each length. Sheet11: Normalized

counts of sRNAseq and total RNAseq reads mapping to the persistently present viruses.
(XLSX)

**S3 Table. Summary of DGE analysis.** The results of DGE analysis using the total RNAseq libraries by the Salmon-DESeq2 pipeline are summarized. Sheet1: Ago-16 KD vs GFP control, Sheet2 RdRP1 KD vs GFP control, Sheet3: RdRP3 KD vs GFP control, Sheet4: TPM values for individual libraries.
(XLSX)

**S4 Table. Summary of expression analysis for repeats.** The results of DGE analysis using the total RNAseq libraries by the Salmon-DESeq2 pipeline are summarized. Sheet1: Ago-16 KD vs GFP control, Sheet2 RdRP1 KD vs GFP control, Sheet3: RdRP3 KD vs GFP control, Sheet4: TPM values for individual libraries. These analyses were done using a reference file containing both protein-coding genes and repeats and data only for repeats are shown.
(XLSX)

**S5 Table. Materials used in this study.** The workbook contains the information of oligos (Sheet1), cell lines (Sheet2) and antibodies (Sheet3).
(XLSX)

**S6 Table. List of commonly misregulated genes upon knockdown of RdRPs and viral infection.** Transcriptome data of cells infected with tick-borne viruses are from [95].
(XLSX)

**S1 Data. Open the "TE_cDNA_virus_links.html" file to view the sortable tables of CDS-,TE- and virus-derived sRNAs in the knockdown libraries.** On the Normalized and Relative levels pages, normalized read counts (RPM) and relative levels compared to the control (dsGFP) library were used. TEs with more than 50RPM and Coding Genes with more than 3.5RPM on average in KD libraries are included in this website. For viral sequences, reference sequences were generated by assembling sRNA sequences (See the Analysis of sRNAseq data section in Materials and methods.) and size distributions of sRNA reads mapped to each contig are shown. Reads were first grouped by their 5' nucleotides, and reads of each length were counted. Full tables including TEs and Coding genes with fewer reads are in the "FullTable" folder.
(ZIP)

**S1 File. Contains dot plots and Gene-GO networks visualizing the results of GO enrichment analysis for misregulated genes upon Ago-16/RdRP1/RdRP3 knockdown.** For dot plots, the size and color represent the number of genes in the GO category and the significance of enrichment, respectively, as indicated in the legend. For Gene-GO networks, the linkage between each enriched GO category and the misregulated genes that are involved in the corresponding GO category is depicted. The color of the dots representing the misreguated genes indicates the log2 fold change level. The size of the dots representing the enriched GO categories indicates the number of misregulated genes in the GO category.
(PDF)

**S1 Raw images.**
(TIF)

## Acknowledgments

The authors are grateful to members of the K.O. laboratories in TLL and NAIST for discussion. We thank Dr. Kentaro Yoshii for critically reading the manuscript. We thank ATCC for ISE6 cells and BGI for sequencing.

## Author Contributions

**Conceptualization:** Canran Feng, Katsutomo Okamura.

**Data curation:** Canran Feng, Mandy Yu Theng Lim, Li-Ling Chak, Masami Shiimori, Kosuke Tsuji, Junko Iida, Katsutomo Okamura.

**Formal analysis:** Canran Feng, Kyosuke Torimaru, Mandy Yu Theng Lim, Li-Ling Chak, Masami Shiimori, Tetsuya Tanaka, Katsutomo Okamura.

**Funding acquisition:** Tetsuya Tanaka, Katsutomo Okamura.

**Investigation:** Canran Feng, Mandy Yu Theng Lim, Li-Ling Chak, Masami Shiimori, Tetsuya Tanaka, Katsutomo Okamura.

**Methodology:** Canran Feng, Kyosuke Torimaru, Li-Ling Chak.

**Resources:** Canran Feng, Kyosuke Torimaru, Mandy Yu Theng Lim, Li-Ling Chak, Kosuke Tsuji, Tetsuya Tanaka, Junko Iida, Katsutomo Okamura.

**Supervision:** Masami Shiimori, Tetsuya Tanaka.

**Visualization:** Canran Feng, Masami Shiimori, Katsutomo Okamura.

**Writing – review & editing:** Canran Feng, Kyosuke Torimaru, Mandy Yu Theng Lim, Li-Ling Chak, Masami Shiimori, Kosuke Tsuji, Tetsuya Tanaka, Katsutomo Okamura.

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
