## [Decision Letter · Decision Letter 0]

1 Dec 2022

PONE-D-22-05293A novel eukaryotic RdRP-dependent small RNA pathway represses antiviral immunity by controlling an ERK pathway component in the black-legged tickPLOS ONE

Dear Dr. Okamura,

Thank you for submitting your manuscript to PLOS ONE. We would like to apologize for the unusually long delay in evaluating your manuscript. After careful consideration, we feel that it has merit but does not fully meet PLOS ONE’s publication criteria as it currently stands. Therefore, we invite you to submit a revised version of the manuscript that addresses the points raised during the review process.

As mentioned by both reviewers, there are some minor concerns that need to be addressed before your manuscript can be considered for publication. These are mainly editing issues and lack of details that should be straightforward to address. Please take these comments into account while revising your manuscript.

We look forward to receiving your revised manuscript.

Kind regards,

Sébastien Pfeffer, PhD

Academic Editor

PLOS ONE

Journal Requirements:

“Research in K.O.’s group was supported by the National Research Foundation, Prime Minister’s Office, Singapore under its NRF Fellowship Programme (NRF2011NRF-NRFF001-042), Temasek Life Sciences Laboratory core funding and the JSPS Fund for the Promotion of Joint International Research (Returning Researcher Development Research, 17K20145). Work in the T.T.’s group was supported by Takeda Science Foundation. The content is solely the responsibility of the authors and does not necessarily represent the official views of these agencies.”

“. We thank ATCC for ISE6 cells and BGI for sequencing. Research in K.O.’s group was supported by the National Research Foundation, Prime Minister’s Office, Singapore under its NRF Fellowship Programme (NRF2011NRF-NRFF001-042), Temasek Life Sciences Laboratory core funding and the JSPS Fund for the Promotion of Joint International Research (Returning Researcher Development Research, 17K20145). Work in the T.T.’s group was supported by Takeda Science Foundation.”

“Research in K.O.’s group was supported by the National Research Foundation, Prime Minister’s Office, Singapore under its NRF Fellowship Programme (NRF2011NRF-NRFF001-042), Temasek Life Sciences Laboratory core funding and the JSPS Fund for the Promotion of Joint International Research (Returning Researcher Development Research, 17K20145). Work in the T.T.’s group was supported by Takeda Science Foundation. The content is solely the responsibility of the authors and does not necessarily represent the official views of these agencies.”

4. PLOS ONE now requires that authors provide the original uncropped and unadjusted images underlying all blot or gel results reported in a submission’s figures or Supporting Information files. This policy and the journal’s other requirements for blot/gel reporting and figure preparation are described in detail at https://journals.plos.org/plosone/s/figures#loc-blot-and-gel-reporting-requirements and https://journals.plos.org/plosone/s/figures#loc-preparing-figures-from-image-files. When you submit your revised manuscript, please ensure that your figures adhere fully to these guidelines and provide the original underlying images for all blot or gel data reported in your submission. See the following link for instructions on providing the original image data: https://journals.plos.org/plosone/s/figures#loc-original-images-for-blots-and-gels. In your cover letter, please note whether your blot/gel image data are in Supporting Information or posted at a public data repository, provide the repository URL if relevant, and provide specific details as to which raw blot/gel images, if any, are not available. Email us at plosone@plos.org if you have any questions.

Reviewers' comments:

Reviewer's Responses to Questions

**Comments to the Author**

1. Is the manuscript technically sound, and do the data support the conclusions?

Reviewer #1: Yes

Reviewer #2: Yes

2. Has the statistical analysis been performed appropriately and rigorously? 

Reviewer #1: Yes

Reviewer #2: Yes

3. Have the authors made all data underlying the findings in their manuscript fully available?

Reviewer #1: Yes

Reviewer #2: Yes

4. Is the manuscript presented in an intelligible fashion and written in standard English?

Reviewer #1: Yes

Reviewer #2: No

5. Review Comments to the Author

Reviewer #1: The authors of this paper experimentally sought to test that RdRPs are directly involved in the production of sRNAs in the model black-legged tick ISE6 cell lines, and that sRNAs control viral transcripts through RNAi and regulation of the Dsor1 gene of the ERK pathway, which contains the 3'UTR of the RdRP-dependent target of repeat-derived sRNAs, and that knockdown of RdRP leads to down-regulation of viral transcripts in a Dsor1-dependent pathway. The authors therefore describe that black-legged tick RdRPs play a role in the biogenesis of specific sRNAs, and in gene regulation and control of viral transcript levels. Complementing the experimental studies on the involvement of RdRPs in RNA silencing in animals other than nematodes (in ticks), experimental evidence for a function in sRNA biogenesis is provided. Several of the following points may require further clarification and interpretation before publication.

1. The authors named the ISE6 cell lines RdRP as “IscRdRP1,3 and 4”, but thereafter, these RdRP names were shown as RdRP1, 3 and 4 throughout the manuscript. Please correct the names and make them consistently.

2. The name format is not consistent: in 1A are “Isc-Ago16” and “Isc-Ago30”; in 1C are “Ago16” and “Ago30”; in 1D are “Ago-16” and “Ago-30”; in the manuscript content are “IscAgo-16” and “IscAgo-30”. Same issue happens to other Ago, RdRp and Aub names. Please keep using one format throughout the manuscript and Figures.

3. In the section of sRNAs produced from repeats, for the description “Large overlaps were seen with Ago-16-RdRP3 and Ago-30-RdRP1 combinations, suggesting that the AGOs and RdRPs might work together to produce repeat associated sRNAs. Interestingly, very few loci overlapped between the three groups, similar to the observation with the sRNAs from coding genes (Figure3)”, however, no Ago-30 knockdown related data is shown in figure 3.

4. As Figure 4 showed that many genes were dependent on the RdRP1-Ago-30, but why RdRP1-Ago-30 combination is not reflected and mentioned in Figure 3A.

5. Quotation mark errors in Figure 6A caption, “Within its 3’UTR, there is a high peak of sRNAs(‘sRNA peak’’)”, please correct the (‘sRNA peak’’) to (“sRNA peak”).

6. Missing supportive evidence to prove that the downregulation of viral gene expression had no difference between Dsor1 single knockdown and Dsor1&RdRP1 knockdown together. The author should introduce a vector containing functional RdRP1 into Dsor1&RdRP1-knockdown cell, and then compare this recovered gene expression with Dsor1 single knockdown to clarify if the result still consistent.

7. The qPCR primer used to amplify Ago-96 cDNA, that noted in Figure S3A, should be listed in Table S5.

Reviewer #2: In this study, Feng et al. identified new factors (RdRP1, RdRP3) and homologs of known factors (Ago3 and Aub) involved in sRNA regulation in Ixodes scapularis ticks and further characterized their role in parallel with other argonaute-like genes previously identified (Ago-16, -30, -78, -96), highlighting synergy and specificity between each factor. The authors further characterize the role of RdRP1 in the regulation of MAPK Dsor1 involved in the antiviral immune response. Overall, they provide a nice, comprehensive and broad analysis of small RNA pathways in Ixodes ticks, through an extensive amount of work, providing valuable data and insights into tick sRNA biology to the scientific community.

While the paper is overall scientifically sound, some minor revisions are required and the author should significantly edit the manuscript to facilitate readability (especially in the abstract and introduction), clarify some elements and reflect the quality of the work performed. See comments for improvement below.

1. The manuscript should have line numbering to properly refer to the text.

2. Authors frequently refer to Figure ‘X’ and Supplementary Data without specifying which supplementary data they are referring to. Please clarify.

3. Authors included description of the data in figure legends. Example Fig 2A ‘Note the dramatic decrease of the “RNAP III” group’. Figure legends should generally only ‘provide a description of the figure that will allow readers to understand it without referring to the text’.

4. Authors provided supplementary information in the form of a 19 pages ‘supplementary PDF’ without describing much of its content. Authors should selectively format it with clear legends and reference clearly in the text.

5. In the abstract, the authors should also avoid generalizing the role of RdRPs as they have shown that RdRP1 and 3 to display different behavior. “RdRP-dependent sRNAs […] are mainly derived from RNA polymerase III-transcribed genes and repetitive elements” is misleading. While virtually all RNAPIII-derived sRNAs appear to be RdRP1-dependent, the regulation of repeat/transposable elements by RdRPs is only punctual and without overlap. Similarly in the second half of the abstract, only RdRP1 was shown to regulate Dsor1 and have antiviral activity, not RdRP3. Please correct throughout the manuscript. For example, in the discussion, page 14 “RdRPs might physically interact with RNAP III during transcription”.

6. Author summary, first sentence. “RdRPs […], but THEIR general importance […].”

7. Introduction, first sentence. Transposable elements cannot be considered foreign nucleic acids. They were integrated in the genome and are now part of the host. Please correct.

8. Introduction, second sentence. Cells do not use small RNAs to distinguish host/foreign nucleic acids. Please correct.

9. The relatively long description of CRISPR in the introduction is questionable as the paper focused solely on eucaryotes and never uses CRISPR technology. It should be shortened or removed.

10. Introduction, page 4, please clarify “worm-like secondary sRNAs”.

11. The authors should discuss the RdRP Ego-1 and other RBPs found by Kurscheid et al, 2009 (https://doi.org/10.1186/1471-2199-10-26).

12. Authors chose to express recombinant tick proteins in mammalian cells, which they acknowledge to have limitations. The authors mention page 20 ‘tick plasmids’, could they clarify? Do they have tick expression plasmids? Alternatively, authors could directly transfect mRNA into tick cells to circumvent the lack of expression vector.

13. In figure 2, sRNAs are shown as a fraction of a whole. Could the authors provide an absolute quantification, ideally normalized to a stable sRNA, for example the SRP RNA shown to be stable across all knockdowns Fig S5B?

14. Fig. 2E-F panels are unrelated to rest of Fig2 and its title. Those data could be associated with the rest of the data related to H. longicornis in Fig S6?

15. Fig. S4D, Ago3-2 knockdown induces a reduction of piRNA of about 50% compared to Ago3-1 knockdown. Also, the authors mentioned that both Ago3-1 and Ago3-2 have ‘very similar sequences’ without specifying percentage of identity. The authors should discuss this difference and if possible, investigate further? Are they both regulating the same piRNA?

16. Fig. S4D, the quantification of antisense piRNA in equivalent amount to sense piRNA is surprising, as piRNA are expected to be derived from the processing of single-stranded RNA. Could it be an artefact? This should be discussed. The author could check for 1U-/10A bias.

17. Page 11, “To test if repeat-associated sRNAs silence expression of repeats”, please rephrase.

18. Figure legends should be homogenous throughout the paper. For example, Fig 2B ‘Read count (RPM)’, 2F ‘Number of reads (RPM)’ and S4 “Normalized read count (RPM)”.

19. Fig. 3A and 4A are not clear. What are the ‘set size’ referring to? The intersection size could be expressed as a percent of effected genes? Also, in Fig. 3B and 4B, does the color code refers to genes regulated by both or either protein? For example, Fig3B shows 5 genes with red histograms, but Fig 3A denotes only 3 genes regulated by RdRP3/Ago-16.

20. Fig. 5B refers to ‘ISCI005428’ while it was stated that the gene is referred to as Dsor1 through the paper. Also, Dsor1 is sometimes referred to as IscDsor1 (discussion, page 15). Please use consistent nomenclature.

21. Description of both qPCR and method referred to as ‘sensor assay’ are missing from the material and methods.

22. The authors should clarify how the sensor assay is normalized. Is Dsor1 3’UTR destabilizing Fluc expression? RdRP1 knockdown only recovers ~20% of Fluc levels compared to the empty control? What is the efficiency of RdRP1 knockdown?

23. Knockdown experiments in Fig. 6-7 should be supported with controls of efficient knockdown similarly to Fig S2 A.

24. Distribution plots in Fig 2C/E, 6A and 7A + supplementary figures are missing a Y-axis legend. Please correct. It would also be best to use consistent scale and different colors for sense and antisense RNA.

25. Author should provide distribution plots of siRNA mapping to each virus mentioned in Fig 7. The authors do not show sufficient evidence to state that there are no virus-derived piRNA. In Fig 7B, they should either use a log scale or split y-axis as the scale only allow to observe the strong antiviral siRNA response and which may hide some processing of viral RNA by the piRNA machinery. This should also be done for each virus as the host response may change for each virus, as shown in Fig. 7C.

26. In the references, ref 71 does not appears to be linked to pEGFP cloning where it is mentioned in page 17. Also, please check conformity of reference formatting with PLOS policy. NCBI links are provided instead of DOI in several cases (ref. 19, 71, …).

6. PLOS authors have the option to publish the peer review history of their article (what does this mean?). If published, this will include your full peer review and any attached files.

Reviewer #1: No

Reviewer #2: No

---

## [Author Response · Author response to Decision Letter 0]

12 Jan 2023

We would like to thank the reviewers for useful comments. We were glad that both reviewers are quite positive. We tried to address all points that were raised by the reviewers either by additional analyses or by editing the text. Below is our point-by-point responses where the comments of the reviewers are in blue italicized letters. 

Reviewer #1: The authors of this paper experimentally sought to test that RdRPs are directly involved in the production of sRNAs in the model black-legged tick ISE6 cell lines, and that sRNAs control viral transcripts through RNAi and regulation of the Dsor1 gene of the ERK pathway, which contains the 3'UTR of the RdRP-dependent target of repeat-derived sRNAs, and that knockdown of RdRP leads to down-regulation of viral transcripts in a Dsor1-dependent pathway. The authors therefore describe that black-legged tick RdRPs play a role in the biogenesis of specific sRNAs, and in gene regulation and control of viral transcript levels. Complementing the experimental studies on the involvement of RdRPs in RNA silencing in animals other than nematodes (in ticks), experimental evidence for a function in sRNA biogenesis is provided. Several of the following points may require further clarification and interpretation before publication.

1. The authors named the ISE6 cell lines RdRP as "IscRdRP1,3 and 4", but thereafter, these RdRP names were shown as RdRP1, 3 and 4 throughout the manuscript. Please correct the names and make them consistently.

We thank the Reviewer#1 for noticing this and we have corrected the names "IscRdRP1,3 and 4" to be RdRP1, 3 and 4 throughout the manuscript.

2. The name format is not consistent: in 1A are "Isc-Ago16" and "Isc-Ago30"; in 1C are "Ago16" and "Ago30"; in 1D are "Ago-16" and "Ago-30"; in the manuscript content are "IscAgo-16" and "IscAgo-30". Same issue happens to other Ago, RdRp and Aub names. Please keep using one format throughout the manuscript and Figures.

We have corrected the names throughout the manuscript and figures except for figure 1A. Because figure 1A shows the result of Phylogenetic analysis involving several species, letter codes such as "Isc-" before the gene names are added to distinguish the species.

3. In the section of sRNAs produced from repeats, for the description "Large overlaps were seen with Ago-16-RdRP3 and Ago-30-RdRP1 combinations, suggesting that the AGOs and RdRPs might work together to produce repeat associated sRNAs. Interestingly, very few loci overlapped between the three groups, similar to the observation with the sRNAs from coding genes (Figure3)", however, no Ago-30 knockdown related data is shown in figure 3.

We removed the erroneous description regarding overlaps between sRNA-producing coding regions that are dependent on Ago-30, as we have no Ago-30 dependent sRNA loci for coding genes based on our cutoffs. We thank the reviewer for pointing this out. 

4. As Figure 4 showed that many genes were dependent on the RdRP1-Ago-30, but why RdRP1-Ago-30 combination is not reflected and mentioned in Figure 3A.

As described in our response above, the descriptions regarding Figure 3A were incorrect, and we removed the sentence.

5. Quotation mark errors in Figure 6A caption, "Within its 3'UTR, there is a high peak of sRNAs('sRNA peak'')", please correct the ('sRNA peak'') to ("sRNA peak").

We thank the Reviewer#1 for noticing this. We have corrected it.

6. Missing supportive evidence to prove that the downregulation of viral gene expression had no difference between Dsor1 single knockdown and Dsor1&RdRP1 knockdown together. The author should introduce a vector containing functional RdRP1 into Dsor1&RdRP1-knockdown cell, and then compare this recovered gene expression with Dsor1 single knockdown to clarify if the result still consistent.

This is an important suggestion, but due to technical limitations, we respectfully decided not to perform such experiments at this point. In ISE6 cells, the transfection efficiency is very low in our hands (about 0.1%) and currently we only have been able to confirm expression of reporter genes (such as luciferases and fluorescent proteins). Rescue experiments with such a low percentage of transfected cells would produce inconclusive results. We are currently working to improve stable transfection methods such that we will be able to perform tests like those suggested by this reviewer in the future. In addition, the results of the suggested experiments may be difficult to interpret because we do not have means to precisely control the levels of RdRP1 expression from the plasmid. An excess of RdRP1 expression itself might affect viral gene expression indirectly, for example.

7. The qPCR primer used to amplify Ago-96 cDNA, that noted in Figure S3A, should be listed in Table S5.

We added qPCR primers of Ago-96 (IscAGO14_690F, IscAGO14_786R) to Table S5.

Reviewer #2: In this study, Feng et al. identified new factors (RdRP1, RdRP3) and homologs of known factors (Ago3 and Aub) involved in sRNA regulation in Ixodes scapularis ticks and further characterized their role in parallel with other argonaute-like genes previously identified (Ago-16, -30, -78, -96), highlighting synergy and specificity between each factor. The authors further characterize the role of RdRP1 in the regulation of MAPK Dsor1 involved in the antiviral immune response. Overall, they provide a nice, comprehensive and broad analysis of small RNA pathways in Ixodes ticks, through an extensive amount of work, providing valuable data and insights into tick sRNA biology to the scientific community.

While the paper is overall scientifically sound, some minor revisions are required and the author should significantly edit the manuscript to facilitate readability (especially in the abstract and introduction), clarify some elements and reflect the quality of the work performed. See comments for improvement below.

1. The manuscript should have line numbering to properly refer to the text.

We have added line numbering in the manuscript.

2. Authors frequently refer to Figure 'X' and Supplementary Data without specifying which supplementary data they are referring to. Please clarify.

We have specified the reference supplementary data in the text according to the Reviewer’s comment.

3. Authors included description of the data in figure legends. Example Fig 2A 'Note the dramatic decrease of the "RNAP III" group'. Figure legends should generally only 'provide a description of the figure that will allow readers to understand it without referring to the text'.

We have moved interpretation of the data in figure 2A legend to the main text (line 213).

4. Authors provided supplementary information in the form of a 19 pages 'supplementary PDF' without describing much of its content. Authors should selectively format it with clear legends and reference clearly in the text.

We add description of the supplementary PDF in the text (Line 1279).

5. In the abstract, the authors should also avoid generalizing the role of RdRPs as they have shown that RdRP1 and 3 to display different behavior. "RdRP-dependent sRNAs [...] are mainly derived from RNA polymerase III-transcribed genes and repetitive elements" is misleading. While virtually all RNAPIII-derived sRNAs appear to be RdRP1-dependent, the regulation of repeat/transposable elements by RdRPs is only punctual and without overlap. Similarly in the second half of the abstract, only RdRP1 was shown to regulate Dsor1 and have antiviral activity, not RdRP3. Please correct throughout the manuscript. For example, in the discussion, page 14 "RdRPs might physically interact with RNAP III during transcription".

We thank the Reviewer#2 for pointing out this problem. We have corrected it throughout the manuscript.

6. Author summary, first sentence. "RdRPs [...], but THEIR general importance [...]."

We have corrected the grammar error.

7. Introduction, first sentence. Transposable elements cannot be considered foreign nucleic acids. They were integrated in the genome and are now part of the host. Please correct.

We have corrected the description.

8. Introduction, second sentence. Cells do not use small RNAs to distinguish host/foreign nucleic acids. Please correct.

We have corrected the description.

9. The relatively long description of CRISPR in the introduction is questionable as the paper focused solely on eucaryotes and never uses CRISPR technology. It should be shortened or removed.

We have shortened the description (line 66).

10. Introduction, page 4, please clarify “worm-like secondary sRNAs”.

We have added description on worm-like secondary sRNAs (line 101).

11. The authors should discuss the RdRP Ego-1 and other RBPs found by Kurscheid et al, 2009 (https://doi.org/10.1186/1471-2199-10-26).

We have added the reference (Line 106).

12. Authors chose to express recombinant tick proteins in mammalian cells, which they acknowledge to have limitations. The authors mention page 20 'tick plasmids', could they clarify? Do they have tick expression plasmids? Alternatively, authors could directly transfect mRNA into tick cells to circumvent the lack of expression vector.

‘Tick plasmids’ that were mentioned here are those used for the luciferase assays. We removed this acknowledgement statement as the producer of the plasmids was already included as an author. We have been able to clearly detect the expression of reporter genes but the transfection efficiency is still far from enough to perform the suggested experiments. As mentioned in our response to question #6 of reviewer 1, the stable transfection method is under development in our laboratory. Establishing mRNA transfection is a possibility, but development of such an experiment is beyond the scope of this manuscript. We recognize that establishing a feasible technique to overexpress proteins in tick cells is important for us and researchers in the field, and we will continue our effort to establish reliable protocols for such experiments. 

13. In figure 2, sRNAs are shown as a fraction of a whole. Could the authors provide an absolute quantification, ideally normalized to a stable sRNA, for example the SRP RNA shown to be stable across all knockdowns Fig S5B?

Accurately normalizing read counts in sRNAseq experiments is always a challenge. As the reviewer pointed out, we calculated the abundance of each species as a fraction of a whole throughout the manuscript, which may produce inaccurate estimates of their abundance especially when the amount of some abundant species dramatically changes. If possible, it would be appropriate to use some “house-keeping sRNAs” for normalization as suggested by the reviewer. However, such “sRNA” species abundantly present in all tissues such as SRP RNA, rRNAs and, tRNAs are out of the size range of library construction (~20-30nt), and not present in our data. We believe our current normalization scheme produces reasonably accurate estimates, as verified by the consistent results in the Northern blotting experiments (Figure S5B).

14. Fig. 2E-F panels are unrelated to rest of Fig2 and its title. Those data could be associated with the rest of the data related to H. longicornis in Fig S6?

We put Fig. 2E-F panels together with the rest of Fig2 panels to show the deep conservation of sRNA production from RNAP III transcribed genes between the two tick species, which suggests the biological importance of this sRNA class. We think this topic might be important enough for the community to be kept as panels in the main figure. For consistence, we change the title of Fig2 to be "Characterization of endogenous sRNA populations in tick".

15. Fig. S4D, Ago3-2 knockdown induces a reduction of piRNA of about 50% compared to Ago3-1 knockdown. Also, the authors mentioned that both Ago3-1 and Ago3-2 have 'very similar sequences' without specifying percentage of identity. The authors should discuss this difference and if possible, investigate further? Are they both regulating the same piRNA?

We apologized for the misleading description in the text: "This contig was potentially derived from two genomic loci with very similar sequences and the sequences similar to ISCI012408 and ISCI004800 were next to each other at both loci". We have rephrased it as "This contig was potentially derived from two genomic loci with ISCI004800(AGO3-2) sequence matched to its first part and ISCI012408(AGO3-1) sequence matched to its second part ". The sequence alignment summary of Drosophila AGO3, ISCI012408(AGO3-1), ISCI004800 (AGO3-2) and the assembled contig is shown below.

 We performed further analysis to show that the changes in the repeat-associated sRNA expression in the AGO3-1 and AGO3-2 knockdown samples were similar but they were distinct from that of the Aub2 knockdown. We generated scatter plots showing the TE distribution in pairwise comparisons among dsAGO3-1, dsAGO3-2, and dsAub2 libraries using the relative RPM level with respect to the control library. We noticed clear differences in the distribution patterns of TEs between dsAGO3-1 vs. dsAub2, dsAGO3-2 vs. dsAub2 and dsAGO3-1 vs. dsAGO3-2. Compared to the control library, the expression change of 25-30nt sRNAs derived from each TE in dsAGO3-1 library seems to be closely correlated with that in dsAGO3-2 library, suggesting their biogenesis might have undergone similar disruption upon dsAGO3-1 or dsAGO3-2 knockdown. These data about piRNA populations decreasing upon each of the knockdowns are now added as Figure S3F.

16. Fig. S4D, the quantification of antisense piRNA in equivalent amount to sense piRNA is surprising, as piRNA are expected to be derived from the processing of single-stranded RNA. Could it be an artefact? This should be discussed. The author could check for 1U-/10A bias.

We checked the strand bias and the ping-pong signature of 25-30nt sRNAs. There appear to be strong strand biases for most repeats, but some repeats produced the piRNA-sized sRNAs at detectable levels (New Figure S8A). Using the established pipeline (DOI:10.1007/978-1-4939-0931-5_12). We detected a clear ping-pong signature, suggesting that ISE6 produces piRNAs using the ping-pong mechanism (New Figure S8B). 

17. Page 11, "To test if repeat-associated sRNAs silence expression of repeats", please rephrase.

We have rephrased it to be "To test if repeat-associated sRNAs regulate the expression of repeats"

18. Figure legends should be homogenous throughout the paper. For example, Fig 2B 'Read count (RPM)', 2F 'Number of reads (RPM)' and S4 "Normalized read count (RPM)".

We have changed to “Normalized read count (RPM)” throughout the paper.

19. Fig. 3A and 4A are not clear. What are the 'set size' referring to? The intersection size could be expressed as a percent of effected genes? Also, in Fig. 3B and 4B, does the color code refers to genes regulated by both or either protein? For example, Fig3B shows 5 genes with red histograms, but Fig 3A denotes only 3 genes regulated by RdRP3/Ago-16.

We followed the convention of the UpSet plot format (DOI: 10.1038/nmeth.3033) and added a reference. ‘set size” shows the number of sRNA whose reads were reduced in the indicated library. We also clarified the color coding of the bards in Fig3B and 4B, in the legend. Bars in the chart are colored when sRNAs are judged dependent on either of the protein in the group. Therefore, 5 genes are colored red in Fig 3B because 3 genes are dependent on both RdRP3 and Ago-16, and 2 genes were dependent only on Ago-16.

20. Fig. 5B refers to 'ISCI005428' while it was stated that the gene is referred to as Dsor1 through the paper. Also, Dsor1 is sometimes referred to as IscDsor1 (discussion, page 15). Please use consistent nomenclature.

We have change to “Dsor1” through the paper.

21. Description of both qPCR and method referred to as 'sensor assay' are missing from the material and methods.

We added a description of qPCR and sensor assay in materials and methods.

22. The authors should clarify how the sensor assay is normalized. Is Dsor1 3'UTR destabilizing Fluc expression? RdRP1 knockdown only recovers ~20% of Fluc levels compared to the empty control? What is the efficiency of RdRP1 knockdown?

We described how the sensor values were normalized in the newly added Sensor Assay part in Materials and Methods. The ratio between firefly luc and Rluc was normalized to that of the empty sensor Dsor1 in each knockdown sample. We do not expect the Dsor1 sensor levels to completely go back to the empty vector level, because other factors such as miRNAs might regulate Dsor1 expression and RNAi-mediated gene knockdown can never achieve the complete removal of gene activity (see Figure S3A).

23. Knockdown experiments in Fig. 6-7 should be supported with controls of efficient knockdown similarly to Fig S2 A.

We assumed that the knockdown efficiency should be similar when the same dsRNAs were used, therefore used Fig S3A as a representative result to show knockdown efficiency.

24. Distribution plots in Fig 2C/E, 6A and 7A + supplementary figures are missing a Y-axis legend. Please correct. It would also be best to use consistent scale and different colors for sense and antisense RNA.

The distribution plots in Fig 2C/E, 6A, and 7A + supplementary figures are screenshots from the UCSC genome browser, where the Y-axis represents the relative abundance (normalized read counts) in the track. We have added a description of the Y-axis for these figures. Because the expression level in Fig 2C/E, 6A, and 7A + vary in a large range, setting a fixed scale among these loci might hide the mapping pattern of some relatively lowly expressed loci. We think using the auto-scaled axis for each locus should be more appropriate when we didn't aim to have a direct comparison of expression level among these figures.

25. Author should provide distribution plots of siRNA mapping to each virus mentioned in Fig 7. The authors do not show sufficient evidence to state that there are no virus-derived piRNA. In Fig 7B, they should either use a log scale or split y-axis as the scale only allow to observe the strong antiviral siRNA response and which may hide some processing of viral RNA by the piRNA machinery. This should also be done for each virus as the host response may change for each virus, as shown in Fig. 7C.

We performed additional analyses to support the conclusion regarding viral piRNAs but no clear evidence for viral piRNAs was found. As a more comprehensive approach, we de novo assembled viral contigs using the small RNA data according to a previously described approach (https://doi.org/10.1093/nar/gkv587, https://doi.org/10.1016/j.virol.2016.10.017). The assembled contigs were searched against the NCBI non-redundant databases and the virus-hit contigs are then used as a reference for mapping the sRNA reads from the control library. Only very minor 25-30nt virus-derived reads could be detected, compared to the abundant 18-24nt sRNA species (New Figure S10A). There is also no significant ping-pong signal for these 25-30nt viral-derived reads (New Figure S10B), in contrast to the clear piRNA signals found at repetitive loci (New Figure S4). 

26. In the references, ref 71 does not appears to be linked to pEGFP cloning where it is mentioned in page 17. Also, please check conformity of reference formatting with PLOS policy. NCBI links are provided instead of DOI in several cases (ref. 19, 71, ...).

Thank you for pointing out this error. We replaced this with a reference that described the modified cloning site of the pEGFP plasmid. And we made the reference format consistent.

---

## [Editor Report · Decision Letter 1]

18 Jan 2023

A novel eukaryotic RdRP-dependent small RNA pathway represses antiviral immunity by controlling an ERK pathway component in the black-legged tick

PONE-D-22-05293R1

Dear Dr. Okamura,

We’re pleased to inform you that your manuscript has been judged scientifically suitable for publication and will be formally accepted for publication once it meets all outstanding technical requirements.

Kind regards,

Sébastien Pfeffer, PhD

Academic Editor

PLOS ONE
---

## [Editor Report · Acceptance letter]

16 Feb 2023

PONE-D-22-05293R1 

A novel eukaryotic RdRP-dependent small RNA pathway represses antiviral immunity by controlling an ERK pathway component in the black-legged tick 

Dear Dr. Okamura:

I'm pleased to inform you that your manuscript has been deemed suitable for publication in PLOS ONE. Congratulations! Your manuscript is now with our production department. 

Kind regards, 

on behalf of

Dr. Sébastien Pfeffer 

Academic Editor

PLOS ONE